



# Colombian soil texture: Building a spatial ensemble model

Viviana Marcela Varón-Ramírez[1], Gustavo Alfonso Araujo-Carrillo[1], and Mario Guevara[2]

[1]Corporación Colombiana de Investigación Agropecuaria - AGROSAVIA, Mosquera-Cundinamarca, Colombia
[2]Centro de Geociencias - Universidad Nacional Autónoma de México Campus Juriquilla, Qro. MX.

**Correspondence:** Viviana Marcela Varón-Ramírez (vvaron@agrosavia.co)

**Abstract.** Texture is a fundamental soil property for multiple applications in environmental and earth sciences. Knowing its spatial distribution allows for a better understanding of the response of soil conditions to changes in the environment, such as land use. This paper describes the technical development of Colombia´s first texture maps, obtained via spatial ensemble of national and global digital soil mapping products. This work compiles a new database with 4203 soil profiles, which were

harmonized at five standard depths (5, 15, 30, 60, and 100 cm) and standardized with additive log-ratio (ALR) transformation. A stack with 83 covariates was developed, including both quantitative and qualitative covariates, and harmonized at 1 square kilometer of spatial resolution. The top explanatory covariates were selected for each transformation in all standard depths through a recursive feature elimination. Ensemble Machine Learning (EML) algorithms (MACHISPLIN and landmap) were trained to predict the distribution of soil particle fractions (SPF) (clay, sand, and silt), and a comparison with SoilGrids (SG)

products was performed. Finally, a spatial ensemble function was created, which identified the fewest prediction errors between EML and SG, then selected the better of these algorithms for each pixel and standard depth. The results of EML algorithms show that the accuracy of MACHISPLIN and landmap were very similar in the SPF at each standard depth, and both were more accurate than SG. The amount of variance explained (AVE) was between 0.12 and 0.35 for EML, and -0.17 and -0.01 for SG; the concordance correlation coefficient (CCC) was between 0.32 and 0.54 for EML, and 0.04 and 0.16 for SG. The best

EML performance was found for the two superficial layers (5 and 15 cm). The accuracy of the spatial ensemble was higher compared to the other algorithms at all standard depths, but the largest improvement was found at the first layer, where AVE values increased between 0.04 and 0.13, and CCC values between 0.04 and 0.10. EML predictions were frequently selected for each PSF and depth in the total area; however, SG predictions were better when increasing soil depth in some specific regions such as Orinoquía and Amazonía. The final error distribution in the study area showed that sand fraction presented

higher absolute error values than clay and silt fractions, specifically in eastern Colombia. The spatial distribution of soil texture in Colombia is a potential tool to provide information for water related applications, ecosystem services and agricultural and crop modeling. However, some aspects must be attended in future efforts to accurately map soil texture; for example, the treatment of abrupt changes in the texture between depths, unbalanced data, and compositional data consistency in spatial ensemble products. Our results and the compiled database (Varón-Ramírez and Araujo-Carrillo, 2021; Varón-Ramírez et al.,

2021) provide new insights to solve some of the aforementioned issues.

    Keywords: Soil Particle Fractions, Ensemble Machine Learning, Compositional Data, Soil Database.



# 1 Introduction

Soil texture is a complex variable characterized by the amount of clay, sand and silt size particles forming soil aggregates. The spatial variability of soil properties and functions is a key component of soil science research, because it allows to understand the response of soil conditions to global environmental or land use changes. Soil textural properties (e.g., particle size soil fraction and distribution) are specifically important to understand soil processes related to agriculture and the environment from the field to the continental scale (Radočaj et al., 2020; Malone et al., 2021; Bönecke et al., 2021; Caubet et al., 2019). For example, soil texture is a basic soil variable for characterizing soil productivity and soil fertility (Patel et al., 2021; Soropa et al., 2021). Knowledge of soil texture spatial variability is also relevant for assessing soil health and developing strategies to reverse land degradation. Soil texture plays a fundamental role in quantifying the capacity of soils to store carbon and to retain the water required for plants to grow (Dharumarajan and Hegde, 2020; Zhang and Hartemink, 2021). Soil texture is directly measured by soil scientists collecting soil samples and performing soil textural analysis in the field or in the laboratory. One common challenge for soil scientists or pedometricians is to accurately predict soil texture (and other soil properties) across areas where no field samples have been collected in the past or at the global scale. At the national to continental scales, there is large uncertainty of current soil texture estimates mainly across large regions and countries of the world where obtaining field soil samples to represent the spatial variability of soil texture is difficult for multiple reasons such as limited funding or site inaccessibility.

Soil texture estimates are available in different formats such as point data, polygon maps or digital soil maps (Poggio et al., 2021; Anderson et al., 2006; Batjes et al., 2017). Spatial predictions of soil properties (e.g., clay content) or classes (e.g., soil textural class) across areas where no soil data exists is the main motivation of digital soil mapping (or pedometric mapping) (McBratney et al., 2003). In digital soil mapping, soil texture for a specific soil depth and for a given location in the geographical space can be predicted as a function (e.g., empirical function) of the soil forming or weathering environment (climate, organisms, topography, geology, ecology, atmosphere and human interventions to soils) (Grunwald et al., 2011). The available soil data collected in the field (e.g., content of sand, silt and clay particles) is used to train algorithms or models in combination with gridded environmental information representing the soil weathering environment. These layers of environmental information are then used as prediction factors for soil property data (continuous or categorical) extracted from soil samples collected at field. These environmental prediction factors are commonly acquired from four main sources: remote sensing, digital terrain analysis, climate, and thematic maps (e.g., soil type, rock type). The use of prediction algorithms or models that are able to account for the spatial variability of soil distribution is the basis of digital soil mapping, while the spatial support of prediction factors representing the soil forming environment is the basis for generating digital soil maps.

Digital soil maps are digital soil datasets that represent the continuous nature of soil variability. Predictions of quantitative soil properties or the probability of presence/absence of a soil class (e.g., a soil textural class) are represented on digital soil maps for a given soil depth and for a specific period of time. These predictions or probability estimates are derived from the





use of statistical models for supervised (at the presence of training data for a response variable) or unsupervised statistical
learning (in the absence of a response variable) (James et al., 2013). Statistical learning methods for supervised learning
(e.g., for upscaling soil texture data using digital elevation models) can be applied to categorical (e.g., to solve classification
problems) or numerical (to solve prediction problems) datasets (Bischl et al., 2016). Currently, there are literally hundreds (if
not thousands) of modeling approaches for solving regression and classification problems. We could classify these methods
in two main modeling cultures: one assumes that the data are generated by a given stochastic data model and the other uses
algorithmic models and treats the data mechanism as unknown (Breiman, 2001). However, it is difficult to classify the large
diversity of modeling approaches and their possible combinations and therefore the problem of model or algorithm selection
to perform predictions in digital soil mapping is an emergent research question.

Geostatistics and machine learning are the two main forms of statistical learning in digital soil mapping. Geostatistics is a
branch of statistics that deals with the values associated with spatial or spatial temporal datasets, whereas machine learning is
a computer-assisted branch of statistics that uses algorithms developed to solve prediction problems. While geostatistics relies
on multiple assumptions about the spatial variability of the target variable, machine learning algorithms could be considered as
assumption-free models. Machine learning models are commonly parameterized (selection of multiple modeling parameters)
using multiple re-sampling techniques, such as cross validation or bootstrapping. These resampling techniques allow the algo-
rithm to 'learn' from the data using the capacity of computers to store results from multiple data configurations following the
same statistical treatment. This computer-assisted learning allows machine learning algorithms to reproduce the relationship
between the response and the prediction factors in the statistical space, and can be applied to soil datasets to generate digital
soil maps. Geostatistics has historically been the main statistical approach to the prediction problem of soil spatial variability
and it is key to determining the spatial structure of error estimates on digital soil maps. In recent decades, machine learning has
also become a conventional approach for digital soil mapping of soil properties and classes. Machine learning algorithms can
be roughly divided in four main groups: a) conventional machine learning based on trees, kernels, linear based or probabilistic
algorithms, b) reinforcement learning algorithms, c) deep learning algorithms and e) ensemble learning algorithms. The pro-
cess of extracting information from the use of multiple modeling approaches and combining them to create a better solution for
a given prediction problem. Machine learning and geostatistics are valuable tools to extract new knowledge around the spatial
variability of soil properties (e.g., soil texture) by the means of digital soil mapping. These models and algorithms are rapidly
evolving, with most developments or applications being in the fields of artificial intelligence, pattern recognition and computer
vision. Recent developments on geostatistics and ensemble learning efforts have demonstrated great potential for improving
the accuracy and spatial detail of current estimates of soil functional properties across scales Hengl et al. (2021); Wadoux et al.
(2020); Llamas et al. (2020).

There is a diversity of emerging research questions in digital soil mapping while coupling the available soil data with the
associated prediction factors in the statistical space. Some examples of these questions are: what is the right pixel size? (Hengl,
2006) what are the best models and prediction factors? (Guevara et al., 2020). What is the sensitivity of models to the size
of training data (Ng et al., 2020), or what are the best big-data management strategies for generating high-spatial resolution
maps across large areas (e.g., countries)? (Shangguan et al., 2014) just to mention a few. Here, we compare and test multiple





machine learning approaches applied to predict soil texture datasets and provide a solution to identify the lowest prediction bias of multiple soil texture predictions across the national scale of Colombia.

Models or algorithms for digital soil mapping are evaluated using information criteria and bias indicators that are usually estimated using resampling techniques, such as spatial and nonspatial cross validation (Wadoux et al., 2021). Some of these accuracy indicators are the root mean squared error, the mean absolute error, the explained variance or the concordance correlation between observed and predicted values, among many others. Here, we use accuracy indicators to evaluate the prediction capacity of multiple machine learning algorithms to generate predictions of soil texture across Colombia. Our objective is the development of a digital soil texture dataset across the country. We present a new dataset composed by harmonized measurements of soil texture (n=4203) that we collect from multiple soil surveys. We harmonize the new dataset with gridded environmental prediction factors and compare the accuracy of our predictions against global datasets. Finally, we develop a pixel wise solution to identify the method with lower prediction bias of multiple soil texture predictions as previously suggested (Gavilán-Acuña et al., 2021).

We hypothesized that multiple prediction algorithms are able to capture the spatial variability of soil texture differently, because they treat the data in different ways to solve prediction problems (e.g., using decision boundaries or probability thresholds or hypothesis of the empirical relationship between the response and the prediction factors). Understanding which are the prediction algorithms and approaches yield lower error levels at the pixel level could benefit model selection efforts in digital soil mapping. The digital soil mapping dataset includes: a) point data to calibrate predictions in the form of a regression matrix, b) environmental prediction factors for soil texture and c) digital soil texture maps and their associated uncertainty. Our model predictions improved the accuracy of global available estimates (e.g,. SoilGrids250m). The main implication of this study is the opportunity to determine the geographical areas where our results improve the accuracy of previous estimates. The digital soil texture dataset is publicly available at: doi:10.6073/pasta/6dded07af834834ee21a134b247507fd (Varón-Ramírez and Araujo-Carrillo, 2021). The digital soil texture maps across multiple soil depths are also available without restrictions at: doi:10.6073/pasta/91441b598d4480091cd7f86f5d3762bf (Varón-Ramírez et al., 2021). This work can have positive implications by increasing the quality of, quantity of and access to soil texture information across Colombia.

## 2 Methodology

A total of five major steps were performed by this work: harmonization and transformation of soil data, adjustment and selection of covariates, spatial prediction with different algorithms, validation, and spatial ensemble.

### 2.1 Dataset

A total of 4203 soil profiles were collected from Sistema de Información de Suelos de Latinoamérica y el Caribe - SISLAC, a soil information system developed by FAO (FAO, 2020). Soil particle-size fractions (PSF) such as clay, sand, and silt were collected, including geographical coordinates (EPGS: 4326). The soil data covered five natural regions (geographic division made based on climatic, vegetation, relief and soil classes conditions) and 31 districts of the continental area of Colombia (Fig.

1). The regions were Caribbean in the north, Pacific in the west, Andean in the center (corresponding to the Andes Mountains), Orinoquia in the east, and Amazon in the south (Rangel-Ch and Aguilar, 1995).

## 2.2   Data harmonization and transformation

Dataset quality was reviewed through consistency inspection. The inspection included two rules: PSF sum equal to 100 and no overlapping between horizons. PSF were transformed at five standard depths (5, 15, 30, 60, and 100 cm), following the vertical discretization as specified in the GlobalSoilMap specifications (Arrouays et al., 2014). That transformation was constructed using a quadratic function of depth with equal areas (spline) (Bishop et al., 1999), through the `mpspline` function of the `aqp` package of R version 4.0.3.

PSF are compositional data and therefore require special treatment before spatial prediction. PSF at each profile in standard depth were transformed based on additive log-ratio (ALR) transformation (Aitchison, 1986), instead of other log-ratio transformations such as isometric, centered and symmetry. The properties of the transformations when applied to regionalized compositions were discussed by Pawlowsky-Glahn and Olea (2004). ALR is commonly used for mapping soil PSF (Odeh et al., 2003; Poggio and Gimona, 2017; Wang et al., 2020; Li et al., 2020), preserving information about spatial correlation and

maintaining the compositional aspect of the variables (Lark and Bishop, 2007).

Let $z_i$, $i$=1, 2, 3 (D) represent the clay, sand, and silt fractions, where $D = 3$ is the number of soil particle-size categories. ALR and inverse ALR transformation are defined as:

$$Trans\_i = ln\left(\frac{z_i}{z_D}\right), \quad i = 1,\, 2,\, \ldots,\, D-1 \tag{1}$$

$$z_i = \begin{cases} \frac{exp\,(Trans\_i)}{1+\sum_{j=1}^{D-1} exp\,(Trans\_j)}, & i = 1,\, 2,\, \ldots,\, D-1 \\[2mm] \frac{1}{1+\sum_{j=1}^{D-1} \exp\,(Trans\_j)}, & i = D \end{cases} \tag{2}$$

where $Trans\_i$ is the transformed value of $z_i$ by the ALR transformation. According with Poggio and Gimona (2017), in this study, clay was used as the denominator variable. The ALR transformation was implemented using the `alr` function in `Compositional` package. The predictive results were back-transformed to PSF (clay, sand, and silt) using `alrinv` function.

## 2.3   Soil covariates

A total of 83 environmental covariates were selected to broadly reflect soil forming factors, as described by McBratney et al. (2003):

$$S_a = f(s, c, o, r, p, a, n) \tag{3}$$

**Table 1.** Environmental covariates used in the work

| Symbol | Soil Forming factor | Number of covariates | Description | Source |
|---|---|---|---|---|
| $s$ | Soil | 28 | Soil index (Clay ratio and GSI). Sand and clay mineralogy. Moisture regime. | Google Earth Engine, 2020 Soil map, scale 1:100.000, IGAC (2012) |
| $c$ | Climate | 3 | 1980 – 2011 annual mean precipitation, relative humidity, and average temperature. | Climatological database, 1980 – 2011. IDEAM (2015) |
| $o$ | Organisms | 8 | Land cover categories. NDVI, bands 6 and 7 from Landsat 8. | Corine Land Cover Classification, scale 1:100.000, 2010–2012. IDEAM (2014) Google Earth Engine, 2020 |
| $r$ | Relief | 23 | Digital elevation model (derived parameters). Physiographic landscape and topography. | SRTM mission of 2000, at 90 m. Soil map, scale 1:100.000. IGAC (2012) |
| $p$ | Parental material | 6 | Lithology. | Soil map, scale 1:100.000. IGAC (2012) |
| $a$ | Age | 10 | Order of soils. | Soil map, scale 1:100.000. IGAC (2012) |
| $n$ | Space | 5 | Oblique geographic coordinates. | Møller et al. (2020) |

where a soil attribute $(S_a)$ is a function of other properties of the soil at a point $(s)$, the climate $(c)$, organisms $(o)$, relief $(r)$, parent material $(p)$, age $(a)$, and space $(s)$ (Table **??**). The pixel size of the environmental covariates was adjusted to 1 square
kilometer. Then, a stack of covariates was created for Colombia.

A recursive feature elimination (RFE) was run for each depth and transformation, using the function `rfe` of the `caret` package. The RFE is an algorithm that implements backwards selection of covariates based on predictor importance ranking (Kuhn et al., 2020). The goal was to find a subset of covariates that were used to produce the most accurate model possible. A regression matrix for each depth and transformation was built with the selected covariates, and this allowed extraction of the
covariate values at the coordinates of each soil sample. With the regression matrix the dataset was divided using a bootstrapping technique (Kuhn et al., 2020): a part for model training (75 %) and another independent part for validation purposes (25 %) (Guevara et al., 2018).

## 2.4 Prediction models

The spatial distribution of the PSF at each of the five standard depths was modeled through Ensemble Machine Learning
(EML) algorithms in two R packages: `MACHISPLIN` (Brown, 2021) and `landmap` (Hengl, 2021).EML consists of various approaches based on different methodologies, including stacking methods, averaging methods, bagging, and boosting approaches(Zounemat-Kermani et al., 2021) .





MACHISPLIN evaluates different combinations from six algorithms to predict the input data, weighing and evaluating the fit. The `interp.rast` function of the MACHISPLIN package interpolates noisy multi-variate data through EML using six algorithms: boosted regression trees (BRT), neural networks (NN), generalized additive model (GAM), multivariate adaptive regression splines (MARS), support vector machines (SVM) and random forest (RF). Further, in MACHISPLIN the residuals of the final model are calculated from the full training dataset and these values are interpolated using thin-plate-smoothing splines. This creates a continuous error surface and is used to correct most of the residual errors in the final ensemble model (Brown, 2021).

Landmap applies the stacking ensemble type. Stacking (sometimes called stacked generalization or committee machine approach) learns in parallel, then fits a meta-model to predict ensemble estimates (Zhang and Ma, 2012). The "meta-model" is an additional model that basically combines all individual or "base learners" (Hengl, 2021). The `train.spLearner` function of the landmap package ensembles the following machine learning algorithms: a fast implementation of RF (ranger), extreme gradient boosting (xgboost), support vector machines (ksvm), neural networks (nnet), and generalized linear models (GLM) with Lasso or Elastic Net Regularization (Cross Validated Lambda) (cvglmnet). The landmap package extends functionality of the mlr 'meta-package' (Lang et al., 2019) and is based in super learner. It is a prediction method designed to find the optimal combination of a collection of prediction algorithms and its framework is built on the theory of cross-validation and allows for a general class of prediction algorithms to be considered for the ensemble (Polley and Van der Laan, 2010).

The two packages share some machine learning algorithms: RF, NN, SVM; but other are different: BRT, GAM, and MARS (in the case of MACHISPLIN), and xgboost and cvglmnet (in the case of landmap). There are other differences: MACHISPLIN performs k-fold cross-validation (k=10). The best model will have the lowest residual sum of squares during cross-validation (Brown, 2021). While, landmap computes 5-fold cross-validation and then used to determine the meta-learner. Nevertheless, cross-validation is spatial. Randomly splitting spatial data can lead to training points that are neighbors in space with test points. Due to spatial autocorrelation, test and training dataset would not be independent in this scenario, with the consequence that cross-validation fails to detect a possible overfitting (Lovelace et al., 2019). That situation is performed by landmap, due it blocks some training points based on spatial proximity to prevent from producing bias predictions (Hengl, 2021).

## 2.5 Validation

One aim of the work was to compare the spatial prediction of techniques mentioned in section 2.4 against the products from SoilGrids (SG) Version 2.0 (Poggio et al., 2021). SG layers of the PSF at each standard depth were downloaded and resampled to the same spatial resolution as the environmental covariates. After, external validation was performed in order to assess their quality, based on the prediction error (difference between predicted and observed values). The quantitative statistics used included the mean error (ME), root mean square error (RMSE), amount of variance explained (AVE), and concordance correlation coefficient (CCC). ME measures bias in the prediction and is defined as the population mean of the prediction errors (Yigini et al., 2018). Values close to 0 indicate that the predictions are unbiased. RMSE is a measure of prediction accuracy and a perfect model would have a value $\approx 0$ (Kempen et al., 2012). The AVE measures the fraction of the overall dispersion of the observed values that is explained by the model, with an optimal value of 1 (Samuel-Rosa et al., 2015). Finally, the CCC



measures the level of agreement of predicted values with observed values (relationship 1:1) (Lawrence and Lin, 1989). These metrics were calculated for EML and SG layers.

In the two EML algorithms and comparative dataset (SG), the prediction errors were estimated with the validation data at each depth. The prediction error was calculated at each point as the difference between the observed and the predicted value (Brus et al., 2011). Each prediction error or independent residual was interpolated using ordinary kriging (OK), a widely used geostatistical technique that assumes intrinsic stationarity (Webster and Oliver, 2007). These layers were obtained through the automatic adjustment of the `automap` package (Hiemstra et al., 2009) and showed the approximated error tend of the estimations of each technique and depth.

## 2.6 Spatial ensemble

The spatial ensemble was generated for each depth from the results of the spatial modelling and its errors. The spatial ensemble was a function created that identified the best result for each depth and each pixel. The created function performed a conditional evaluation using the maps of all the techniques and their respective errors, subsequently, for each pixel, it identified which model had the least error and selected it. In this way, a final map was built with the best result including all the created techniques. The quantitative statistics were newly calculated by the final calculated maps.

## 3 Results

The main product of this work is a group of maps that contains the PSF predictions for each standard depths. The predictions were made with EML algorithms, and these predictions were compared with SG products. Additionally, a spatial ensemble (with EML and SG) were constructed for each PSF in five standard depths. Following, the principal process steps are going to be shown in the next subsections.

### 3.1 Soil texture characterization and dataset

The textural classes for each sample point at 5 standard depths are shown in Fig. 2. In most USDA textural classes, the dataset has soil samples. SL, L, CL, SCL and C were the most frequent textural classes in all standard depths, and SI and SIL textural classes, were less frequent. However, the silt-size particles increased in the deepest layers

Descriptive statistics are shown in Table ?? for PSF and its transformations (Trans_1 and Trans_2). In all textural fractions, the minimum contents were 1% or less for 5 standard depths and the maximum contents for silt fractions were less than clay and sand. The mean and median for sand fraction was higher than clay and silt fractions in all standard depths. However, the differences between the median for clay, sand and silt in the deepest standard depths were less than the most superficial. The SD grows for all fractions in the last two depths and sand content was the fraction with highest SD in all depths. Kurtosis coefficients were negative for sand fraction in all depths and for the clay fraction in the last three depths; in contrast, for silt fractions, the kurtosis were always positive. The skewness coefficients were positive and less than 1 for all textural fraction datasets.



**Table 2.** Descriptive statistics of PSF and its transformations for each standard depth. Min: minimum; Max: maximum; SD: standard deviation

| Standard depth (cm) | PSF and transformations | Min | Max | Mean | Median | SD | Kurtosis | Skewness |
|---|---|---|---|---|---|---|---|---|
| 5 | Clay | 0.02 | 95.07 | 27.88 | 25.17 | 17.23 | 0.43 | 0.82 |
| | Sand | 0.13 | 99.19 | 42.69 | 42.00 | 22.45 | -0.83 | 0.13 |
| | Silt | 0.10 | 83.65 | 29.43 | 28.00 | 13.95 | 0.00 | 0.46 |
| | Trans_1 | -6.06 | 7.68 | 0.44 | 0.50 | 1.44 | 1.20 | -0.36 |
| | Trans_2 | -6.33 | 7.87 | 0.14 | 0.15 | 0.89 | 4.19 | -0.22 |
| 15 | Clay | 0.44 | 94.50 | 28.64 | 26.07 | 17.07 | 0.28 | 0.76 |
| | Sand | 0.12 | 98.00 | 42.02 | 41.53 | 22.18 | -0.83 | 0.17 |
| | Silt | 1.00 | 81.74 | 29.33 | 28.00 | 13.61 | 0.00 | 0.47 |
| | Trans_1 | -6.42 | 4.63 | 0.38 | 0.43 | 1.40 | 1.01 | -0.40 |
| | Trans_2 | -3.45 | 0.11 | 0.12 | 4.82 | 0.82 | 1.51 | -0.26 |
| 30 | Clay | 0.29 | 94.78 | 30.47 | 28.40 | 17.75 | -0.15 | 0.58 |
| | Sand | 0.04 | 98.00 | 40.48 | 38.53 | 22.62 | -0.82 | 0.27 |
| | Silt | 0.36 | 76.76 | 29.05 | 27.34 | 13.75 | 0.05 | 0.55 |
| | Trans_1 | -6.83 | 5.83 | 0.27 | 0.27 | 1.45 | 1.05 | -0.32 |
| | Trans_2 | -3.84 | 3.33 | 0.04 | 0.05 | 0.83 | 1.00 | -0.07 |
| 60 | Clay | 0.01 | 94.50 | 32.32 | 30.10 | 19.16 | -0.47 | 0.45 |
| | Sand | 0.03 | 99.86 | 39.10 | 36.08 | 23.67 | -0.79 | 0.38 |
| | Silt | 0.05 | 90.13 | 28.58 | 26.13 | 14.61 | 0.15 | 0.66 |
| | Trans_1 | -7.47 | 9.14 | 0.16 | 0.14 | 1.60 | 1.53 | -0.20 |
| | Trans_2 | -4.67 | 6.78 | -0.03 | -0.05 | 0.93 | 1.84 | 0.21 |
| 100 | Clay | 0.06 | 97.07 | 32.76 | 30.71 | 19.70 | -0.55 | 0.41 |
| | Sand | 0.01 | 99.80 | 38.57 | 34.76 | 24.38 | -0.72 | 0.46 |
| | Silt | 0.14 | 87.50 | 28.67 | 26.24 | 15.40 | 0.33 | 0.71 |
| | Trans_1 | -8.43 | 7.35 | 0.13 | 0.08 | 1.64 | 1.22 | -0.16 |
| | Trans_2 | -4.25 | 4.19 | -0.05 | -0.07 | 0.94 | 1.04 | 0.14 |

## 3.2 Covariate selection

The selection of the best covariate for each transformation was made, in all standard depths, using a recursive feature elimina-
235 tion (rfe) algorithm (Table **??**). Ten rfe models were obtained and individual covariate stacks were built for each transformation
in all standard depths. For each standard depth, covariates were selected for Trans_1 and Trans_2: 44 and 83 (5 cm), 54 and
54 (15 cm), 59 and 83 (30 cm) 56 and 58 (60 cm), and 56 and 83 (100 cm) (Table 3). The Top 5 covariates for each selec-
tion included soil forming factors: climatic (TEM, RH and PPT), topographic (altitude, slope, and presence of Flood Planes),

**Table 3.** Top 5 covariates selection for each transformation and standard depth. TEM: temperature, RH: relative humidity, PPT: precipitation, L8 b7: Landsat 8 band 7, L8 b6: Landsat 8 band 6, GSI: Grain Size Index.

| Standard depth (cm) | Variable | Covs selected | RMSE | Top 5 Covs selected |
|---|---|---|---|---|
| 5 | Trans_1 | 44 | 1.21 | TEM, RH, PPT, Altitude, L8 b7 |
| | Trans_2 | 83 | 0.76 | TEM, RH, PPT, Altitude, Alluvial |
| 15 | Trans_1 | 54 | 1.17 | TEM, Altitude, RH, PPT, Clay ratio |
| | Trans_2 | 54 | 0.74 | TEM, RH, PPT, GSI, Altitude |
| 30 | Trans_1 | 59 | 1.24 | TEM, RH, Altitude, Flood plane, PPT |
| | Trans_2 | 83 | 0.73 | Alluvial, PPT, RH, TEM, Altitude |
| 60 | Trans_1 | 56 | 1.34 | RH, TEM, PPT, Altitude, Slope |
| | Trans_2 | 58 | 0.82 | L8 b6, PPT, Alluvial, RH, Clay ratio |
| 100 | Trans_1 | 56 | 1.34 | PPT, RH, TEM, Altitude, L8 b6 |
| | Trans_2 | 83 | 0.84 | L8 b6, PPT, GSI, Alluvial, Clay ratio |

parent material (presence of Alluvial materials), activity of organisms (bands 6 and 7 of Landsat 8) and previous soil index
information (Clay ratio and SGI). Only two binary covariates (Alluvial and Flood Plains) were represented in this selection.

### 3.3 Soil texture predictions and SG products validation

Boundary adjustment parameters of external validation are given in Table **??**. Referring to ME, the highest values were found
in silt fraction, principally in the 100 cm layer with 3.16% and 2.82% for MACHISPLIN and landmap algorithms, respectively.
Clay and sand fractions for both algorithms showed negative ME values (underestimation), except for clay content in the 100
245  cm layer. Also, clay and sand in the two deepest layers showed the closest to zero ME values for both MACHISPLIN and
landmap. For MACHISPLIN and landmap, the RMSE values were similar for each PSF and depth, with the lowest RMSE
values found at 15 cm.

The AVE values were from 0.12 to 0.35. For all standard depths and algorithms, sand fraction had higher AVE values
than clay and silt, except in the 60 cm layer, where AVE for sand and silt were equal. For MACHISPLIN algorithm, the AVE
decreased with increased depth. In contrast, for landmap algorithm, the behavior of AVE values were not correlated with depth.
On the other hand, the CCC values were from 0.32 to 0.54. The CCC values for sand were higher than clay and silt for all
depths and algorithms, and the lowest CCC value between standard depths was found in the deepest layer (100 cm).

An evaluation of SG (250 m) products with the dataset validation (the same used for EML validation) was made, and is also
shown in Table **??**. Negative ME values (underestimation) were found with a range between -8.28% to -6.21% for sand fraction
and positive values with range between 0.89% to 3.42% and 3.47% to 4.78% for silt and clay, respectively. This ME values
were higher than EML predictions. In respect to RMSE values, sand fraction had higher values than clay and silt fractions and



the RMSE increased with increased depth, which was equal in EML predictions. For AVE, negative and close to zero values were found in all standard depths and fractions (-0.17 to -0.01), which were fewer than EML results. Similarly, the CCC values were close to zero (0.04 to 0.16) and the highest values were obtained for sand and silt fraction in the three most superficial layers.

## 3.4 Spatial ensemble

The Fig. 4, 5, 6, 7, 8 display the final spatial ensemble maps for each PSF, which contain their final error and the model selected at each pixel at each standard depth. The spatial ensemble, which, as described above, is a collection of best-fit data from 3 separate algorithms (MACHISPLIN, landmap, and SG), contained common elements/features in most standard depths.

The MACHISPLIN algorithm (Yellow colors) was the model with less percentages errors of clay at 5, 15, 30 and 100 cm; at 60 cm, this algorithm was the best fit for both sand and clay. MACHISPLIN had representation in all natural regions, and in the deepest layers. These predictions were most commonly chosen in areas that were extensive and continuous, such as Orinoquía and Amazonía region for sand at 60 cm and Caribe region for clay at 100 cm. At 30 cm layer, MACHISPLIS was the poor predictor for silt (less yellow areas).

By the other hand, landmap (green colors) was the algorithm with less percentages errors of sand and silt at all standard depths, except by the 60 cm, where this algorithm was the best predictor of clay. The landmap had representation in all natural regions. for silt at the deepest layers, landmap had areas that were large and continuous; in contrast, for sand, landmap was selected mostly in Orinoquia and Amazonía region.For clay at 5 cm, landmap was the less model selected, and the areas were not continuous.

Concerning to SG (gray colors), with depth raising the SG selection increased, specially at 30 and 100 cm layers. The largest areas that were represented by SG products are mainly in Orinoquía and Amazonía region for sand and silt; In contrast, for clay, SG was chosen in continual areas in the country. This product had the fewest contribution in the most superficial layers.

The external validation showed an improvement in their metrics vs. the use of a single algorithm (Table **??**).RMSE values decreased for all PSF and standard depths, except for MACHISPLIN at 15 cm in silt fraction. By the other hand, AVE values increased with the spatial ensemble model; interesting that the sand fraction had the most improvement respect to clay and silt fraction. Concern to CCC values, these still equal or higher than individual algorithms, except by clay at 5 and 60 cm, and silt at 15 and 100 cm in MACHISPLIN. Finally, concern to ME, these values were fewer than ME values of SG algorithm, however for MACHISPLIN and landmap the behavior of ME increased and decreased randomly with the depth increase.

## 4 Discussion

In this paper we developed a new digital soil texture dataset that contains legacy soil data, environmental covariates and the first digital soil texture maps across Colombia. Soil texture is a key property required for many applications in environmental sciences. Colombia's literature on machine learning applied to soil texture mapping is limited. Our results are based on state of the art ensemble learning. We improve the accuracy and detail of previous conventional previous maps. While many studies





**Table 4.** Adjustment parameters of each algorithm for PSF in five standard depths. ME: Mean Error; RMSE: Root Square Mean Error; AVE: Amount of variance Explained; CCC: Concordance Correlation Coefficient

| Depht (cm) | Method | MACHISPLIN | | | Landmap | | | SoilGrids | | |
|---|---|---|---|---|---|---|---|---|---|---|
| | PSF | Clay | Sand | Silt | Clay | Sand | Silt | Clay | Sand | Silt |
| 5 | ME | -1.00 | -0.71 | 1.71 | -1.88 | -0.06 | 1.94 | 3.68 | -7.82 | 3.42 |
| | RMSE | 15.13 | 18.61 | 12.07 | 15.03 | 18.53 | 12.11 | 18.24 | 23.84 | 14.37 |
| | AVE | 0.26 | 0.34 | 0.26 | 0.27 | 0.35 | 0.26 | -0.07 | -0.07 | -0.05 |
| | CCC | 0.42 | 0.51 | 0.44 | 0.40 | 0.54 | 0.36 | 0.08 | 0.13 | 0.15 |
| 15 | ME | -0.30 | -0.75 | 1.05 | -0.90 | -0.47 | 1.37 | 3.47 | -7.15 | 2.86 |
| | RMSE | 14.36 | 18.05 | 11.39 | 14.49 | 18.47 | 11.70 | 17.89 | 22.94 | 13.85 |
| | AVE | 0.31 | 0.34 | 0.29 | 0.29 | 0.31 | 0.25 | -0.08 | -0.06 | -0.05 |
| | CCC | 0.46 | 0.52 | 0.46 | 0.43 | 0.54 | 0.36 | 0.08 | 0.13 | 0.13 |
| 30 | ME | -0.50 | -1.09 | 1.59 | -0.76 | -0.47 | 1.23 | 3.62 | -6.95 | 2.42 |
| | RMSE | 15.73 | 19.06 | 11.81 | 15.83 | 18.93 | 11.75 | 18.42 | 23.59 | 14.06 |
| | AVE | 0.23 | 0.31 | 0.29 | 0.22 | 0.32 | 0.29 | -0.06 | -0.05 | -0.01 |
| | CCC | 0.37 | 0.48 | 0.45 | 0.38 | 0.51 | 0.46 | 0.09 | 0.13 | 0.16 |
| 60 | ME | -0.20 | -0.78 | 0.98 | -0.08 | -0.76 | 0.84 | 4.64 | -6.21 | 0.89 |
| | RMSE | 17.11 | 20.74 | 13.11 | 17.08 | 20.87 | 13.30 | 20.35 | 24.55 | 15.39 |
| | AVE | 0.23 | 0.25 | 0.25 | 0.23 | 0.24 | 0.23 | -0.09 | -0.05 | -0.04 |
| | CCC | 0.37 | 0.42 | 0.41 | 0.38 | 0.43 | 0.39 | 0.06 | 0.09 | 0.10 |
| 100 | ME | -0.03 | -3.12 | 3.16 | 0.05 | -2.87 | 2.82 | 4.78 | -8.28 | 2.39 |
| | RMSE | 17.35 | 22.00 | 14.18 | 17.53 | 22.36 | 14.24 | 20.77 | 26.55 | 15.81 |
| | AVE | 0.23 | 0.20 | 0.13 | 0.21 | 0.17 | 0.12 | -0.10 | -0.17 | -0.08 |
| | CCC | 0.35 | 0.38 | 0.32 | 0.37 | 0.38 | 0.34 | 0.06 | 0.04 | 0.08 |

**Table 5.** Adjust parameters of spatial ensemble model by size particle.

| Depth (cm) | 5 | | | 15 | | | 30 | | | 60 | | | 100 | | |
|---|---|---|---|---|---|---|---|---|---|---|---|---|---|---|---|
| PSF | Clay | Sand | Silt | Clay | Sand | Silt | Clay | Sand | Silt | Clay | Sand | Silt | Clay | Sand | Silt |
| ME | 2.54 | -1.02 | 1.68 | -0.36 | -0.70 | 1.02 | -0.16 | -1.11 | 0.96 | -0.01 | -1.22 | 0.62 | 0.11 | -2.12 | 1.68 |
| RMSE | 13.99 | 16.91 | 11.79 | 14.14 | 16.66 | 11.71 | 15.68 | 18.09 | 11.40 | 16.63 | 20.13 | 13.03 | 17.15 | 21.75 | 14.17 |
| AVE | 0.37 | 0.46 | 0.30 | 0.33 | 0.44 | 0.25 | 0.23 | 0.38 | 0.34 | 0.27 | 0.29 | 0.26 | 0.25 | 0.21 | 0.13 |
| CCC | 0.19 | 0.61 | 0.48 | 0.46 | 0.59 | 0.43 | 0.38 | 0.53 | 0.48 | 0.39 | 0.45 | 0.40 | 0.38 | 0.38 | 0.26 |





focus on mapping soil properties such as pH and organic matter, less studies focus on comparing and testing approaches for
maximizing accuracy. We improve the accuracy of national soil texture predictions, with a fully independent dataset, respect
to global products. Also, we provide new insights for assessing the quality and accuracy of global soil texture predictions.
However, we also identify areas where global predictions suggest lower prediction variance. Our results contribute with a
nation-wide benchmark of the reliability of global predictions compared to national predictions. We base our soil texture
predictions in a soil texture dataset with a transformation process. We deliver this soil texture dataset and demonstrate its
applicability using digital soil mapping to describe the geography of Colombia's soil texture. We first discuss the general
geography of soil texture across the country and then we compare and discuss our findings with previous work.

Colombia has a great diversity of soil, which changes with depth. In the 5 standard depths, soil texture in Colombia has
representation in all textural classes defined by Staff (2014). As depth increases, the soil thins, and the proportion of clay and
silt rises. This is evident in southern, southeastern, eastern, and northeastern Colombia, where Fig. 2 demarcates this with
300 redder colors at 60 and 100 cm. On the other hand, coarse soils (blue colors) are in central and northern areas, and these soil
textures hold with increasing depth. This high diversity of soil texture is due to the high number of interactions between soil
forming factors, specifically the great diversity of parent materials, within Colombia (IGAC, 2015; Araujo et al., 2017).

Some topography and parent material covariates were the principal drivers in texture modeling. The key areas with fine
and medium textures are found in the northwest (Floodplain and land depressions), in central areas (Magdalena River valley),
in the west (Cauca River valley), in the south (Amazon region), and in the east (Orinoquia region). All these regions have
a common soil forming factor such as alluvial parent material that is deposited by one or many rivers, that are fine soil
fractions driver (Flórez, 2003). 2003). On the other hand, medium to coarse textures are principally found in the hillsides of
mountain landscapes in central, southern, and southwestern regions. Mainly, these coarse soil textures are due to the presence
of sandstone, conglomerate sandstone, granites and gneisses, among others, that have siliceous and quartz rocks (Catoni et al.,
2016), volcanic materials and glacial clast (IGAC, 2015) that are presented in these areas. Despite the relationship between soil
texture distribution and relief and parent material covariates, only altitude (quantitative), slope (quantitative), alluvial (binary)
and Floodplane (binary) covariates were shown in the top 5 predictors for each standard depth.

Although parental material is very important in the soil texture spatial distribution, the covariates selection identified that
the climatic covariates were more important (i.e., TMED, RH, and PPT). The covariates used to describe the parental material
were binary class variables, maybe following exercises should include quantitative variables to identify this soil forming factor,
for example using radar remote sensing (Niang et al., 2014) or based in the spectral response in the visible and near infrared
spectrum (Vis-NIR), medium infrared (MIR), and Vis-NIR-MIR (Campbell et al., 2019). In the PSF predictions (in specific the
ALR components), the importance of the climatic covariates did not have obvious changes with depth. The climate conditions
of the country have led to relatively strong physical weathering in soil forming process (Osman, 2013). Due to the country's
location, it is influenced climatologically by the atmospheric circulation of the Caribbean Sea, Pacific Ocean, the Amazon
basin, and the orographic barrier of the three branches of the Andes Mountain (Poveda, 2004). Furthermore, in this study the
variables were chosen to maximize the predictive power of the models not their explanatory capabilities.



Colombia has not produced maps on PSF at a national scale with DSM products but is has developed soils surveys through conventional mapping. These maps use a series of delineated polygons based on qualitative soil characteristics called carto-
graphic soil units (CSU), and have been produced in different periods: Cortés et al. (1982) (scale 1:5.000.000), IGAC (2003) (scale 1:500.000), and IGAC (2015) (scale 1:500.000). The map carried out by IGAC (2015) represented the PSF through four textural groups of soils: very fine, fine, medium, and coarse. In that study, each CSU's group texture was calculated whit a weighted average according with each soil profile representation in the CSU and the textural group for each profile was calcu-lated with a weighted average soils fraction between 0 and 50 cm depth according to horizon thickness. The best example of the subnational scale is the work developed by Araujo-Carrillo et al. (2021). In the Cundiboyacense high plateau, their study represented the clay fraction and textural classes only for the surface layer of the soil (0-20 cm) and used the random forest machine learning algorithm.

The textural soil distribution of Colombia presented in this study is not directly comparable with previous national textural soil maps. Due to the methodology used in IGAC (2015), the depth studied is different, the polygons delineated (CSU) have a unique value for an entire area, and CSUs are not an uncertainty value associated. These last two reasons are the major use limitations in traditional soil surveys (Angelini et al., 2016). Despite that, the maps produced by this study and those of the IGAC project, both show 2 major areas with similar attributes. In northwest (Caribbean region) and southern (Amazon region), the IGAC study present a fine group texture (clay between 40 and 60%) and this current result shows that levels of clay percentages in that clay range. However, there are a principal region in western (Orinoquia region), where the two results are very different. The previous result shows these areas with coarse textural group and this current result displays low percentages of sand fractions for 15, 30, and 60 cm depths. These differences are due to the low soil sampling density; where there is just one observation, and in this current study, its nearest predictions are driven by soil data.

In other countries there have been several experiences with the mapping the PSF at different depths using DSM products: France (Mulder et al., 2016), Scotland (Poggio and Gimona, 2017), Hungary (Laborczi et al., 2019), or China (Liu et al., 2020) are some examples. However, few have used the spatial ensemble techniques. One representative case was developed by Hengl et al. (2021) for the continent of Africa at three depths (0, 20 and 50 cm) and at 30 m spatial resolution. They produced predictions using 2 scale 3D EML framework implemented in the `mlr` R package. Their study utilized an improved predictive mapping framework: spatially-adjusted EML, that better accounts for spatial clustering of points. A special point of their work was the spatial cross-validation methodology, obtaining the following RMSE for $\approx 122.200$ training samples: clay 9.6%, sand 13.7%, and silt 8.9%. Their results proved to be more accurate that previous works, which is attributable to the addition of higher resolution remote sensing images and Digital Terrain Parameters (DTM), the adoption of methodological improvements in hyper-parameter tuning, feature selection, and ensembling of models using the Super Learner algorithm (Hengl et al., 2021). Of the above considerations, we did not work with higher resolution covariates from earth observations data or DTM derivatives, but we implemented several concepts, like spatial cross validation, feature selection, and the use the EML.

Although it is a challenge to compare the results from several studies, especially as the focus of each study is different, only as a reference point, a comparison was developed to obtain statistics with the works from France, Scotland, Hungary, and



China. At topsoil depth, the CCC obtained by Mulder et al. (2016) was similar in the sand (0.63), but more accurate in the clay (0.53) and the silt (0.61). The process employed regression tree modelling (cubist), a data-mining technique that was not used in the EML employed in this work. In the case of Scotland, between 0 to 100 cm depth, the average RMSE was 7.10% in the clay, 15.86% in the sand and 11.44% in the silt. All the statistics were better, and they used the hybrid geostatistical Generalized Additive Models (GAMs), combining GAM with Gaussian simulations. About 9000 sampled locations were available for that model in almost 78.000 km$^2$ (Poggio and Gimona, 2017). Laborczi et al. (2019) produced maps of PSF on Hungary through composite regression kriging directly compiled (D) and synthetized (S), obtaining RMSE in the clay of 8.91% (D) and 9.32% (S), and the sand of 16.38% (D) and 16.92% (S). The results in Hungary were similar in sand that calculated in this work. In China with spatial resolution of 90 m and with 4579 soil profiles, Liu et al. (2020) obtained an RMSE of 9.79% in clay, 18.65% in sand and 14.76% in silt, with statistics very similar at other depths. They constructed random forest models with a limited number of sparse soil profiles sites, i.e., nearly one soil profile site per 2000 km$^2$ on average in their study area. We obtained better results in sand and silt, but worse in clay at the same depth.

In the case of SG 2.0 the RMSE was 13% for clay, 18% for sand and 13% for silt. The qualitative evaluation showed that coarse scale patterns are well reproduced (Poggio et al., 2021). The SG products showed good results in silt in the spatial ensembled generated, but in specific depths (30 cm and 100 cm) and regions (zones east and south of the country). In other PSF and depths, the spatial ensembled showed better results with the EML worked.

According to the description above, the results of the spatial ensembles were acceptable in general terms, and this work could identify the better models, error trends, and prediction layers of the PSF. However, in many areas, depths, and textural fractions the results did not have good quantitative statistics. The causes can be many: the relations between some soil properties and landscape attributes are nonlinear, complex, or unknown, a concept defined by Minasny and McBratney (2010). Linked to the aforementioned is the distribution of the soil samples. The study had an unbalanced representation and spatial clustering, for example the central zone (Andean region) was the most represented (bias towards potentially productive areas), while the east and southeast zones were the least represented (Fig. 1), so many predictions were largely controlled by point data, a similar case to that reported by Hengl et al. (2014). It is important to say that in the SISLAC database some legacy soil profiles had significant positional errors, due to the proposals of traditional soil surveys carried out by the IGAC.

Overall, this study used the best available environmental information, and it represented the PSF for all country at five standard depths. However, the differential factor included maps that represent the best model (EML or SG) in each area of the country at different depths, called in this work spatial ensembled. Nevertheless, the approach has limitations, for example with the results of the spatial ensembled it is not possible to sum 100 with all PSF, because this technique is a combination the best EML or SG worked. Also, there is an abrupt change in the PSF through depths in some areas, a normal behavior in some kinds of soils (for example from alluvial parent material), but not on soils like those found in the peneplain landscape.

In Colombia, DSM has new and great challenges to attend map-user's requirements, such as soil texture predictions with uncertainty improvements. There are two principal strategies to improve predictions: treatment of unbalanced soil-data and incorporation of new environmental-covariates related to soil texture drivers. Attending the first strategies, is necessary to raise the soil data base whit available soil information from other sources such as detailed soil surveys, soil degradation, and soil



management studies made by national and governmental institutions (e.g. IGAC, IDEAM or UPRA); or obtaining soil textural

fractions from other kind of soil analysis, such as Visible Near InfraRed-Short of soil minerals (Lagacherie et al., 2020). Also,

model-building processes by soil group (Kempen et al., 2009) or homosoil (Angelini et al., 2020; Malone et al., 2016) have

been used to get pedologically-plausible predictions in areas without high soil-sampling density. Finally, taking heed of second

strategies there are some qualitative and quantitative environmental covariates that could buttress the predictors stack such as

depth to bedrock, and soil horizons designations and thickness.

## 5   Conclusions

We provided the first comparison of the PSF across Colombia between EML models (MACHISPLIN and landmap) and the

existing soil texture maps provided by SG. The study shows that the prediction of the spatial distribution of soil texture

with national datasets was better an average 17% (in terms of RMSE) using EML models than the SG products. Between

MACHISPLIN and landmap there was no better EML model, because the quantitative statistics were very similar. In function

of the PSF, the spatial distributions did not exhibit a fraction with better results. While the silt had the lowest RMSE at different

depths, the sand had the highest AVE, and the clay had the lowest ME. However, in the case of the depths, at 5, 15 and 30

cm were obtained the better results for all the PSF, while at 60 and 100 cm the worst, which indicates the effectiveness in the

depths closest to the soil surface.

Another valuable contribution developed in this study was the implementation of the spatial ensembled of soil texture

fraction on a national scale and at different depths. This implementation identified the best result for each depth and each pixel.

Although the SG products had the worst quantitative statistics overall, in some areas of the country these products performed

well, such as with 30 cm silt in the south. However, with the spatial ensembled was possible to get the best composition of the

models worked.

The prediction of the spatial distribution of soil texture fraction obtained can provide soil information for water related ap-

plications, ecosystem services, agricultural and crop modeling. However, the results had limitations, especially with the abrupt

change in the texture fraction through the depths in some areas and the handling of the compositional data (sum equal to 100)

in the spatial ensembled products. Treatment of unbalanced soil-data and incorporation of more appropriate environmental-

covariates are key to improving accuracy in the future.

## 6   Data availability

Dataset are available at: https://portal.edirepository.org/nis/mapbrowse?packageid=edi.746.1. This repository contains the data

set for each standard depth. For each sample point are shown PSF and ALR transformations (Trans_1 and Trans_2) (Varón-

Ramírez and Araujo-Carrillo, 2021)

Textural soil maps are available at: https://portal.edirepository.org/nis/mapbrowse?packageid=edi.972.1. In this repository

the users are going to find 9 raster stacks: PSF obtained with landmap and MACHISPLIN algorithms (2 stacks); PSF obtained

from SG (1 stack); residual of the PSF predictions for landmap and MACHISPLIN algorithms and SG (3 stacks); and finally

PSF predictions obtained through spatial ensemble technique (3 stacks). All stacks contain information at 5 standard depths (Varón-Ramírez et al., 2021)

*Author contributions.* Varón-Ramírez contributed with conceptualization of soil textural-data management methodology, data cleaning, harmonization of the soil dataset, writing of results and discussion and elaboration of cartography results. Araujo-Carrillo contributed with environmental covariates construction, writing of methodology, results and discussion and elaboration of cartography results. Guevara con-

tributed with conceptualization of spatial ensemble models strategies and the comparison with global products and writing the introduction and discussion.

*Competing interests.* The authors declare that they have no conflict of interest.

*Acknowledgements.* The authors want to thank the Ministerio de Agricultura y Desarrollo Rural (MADR), the Corporación Colombiana de Investigación Agropecuaria (AGROSAVIA),and the Universidad Nacional Autónoma de México (UNAM) for supporting this study. Mario

Guevara acknowledges support by the Programa de Apoyo a Proyectos de Investigación e Innovación Tecnológica (PAPIIT) under grant number IA204522.



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





**Figure 1.** Soil-sample points distribution at 5 standard depths in Colombia



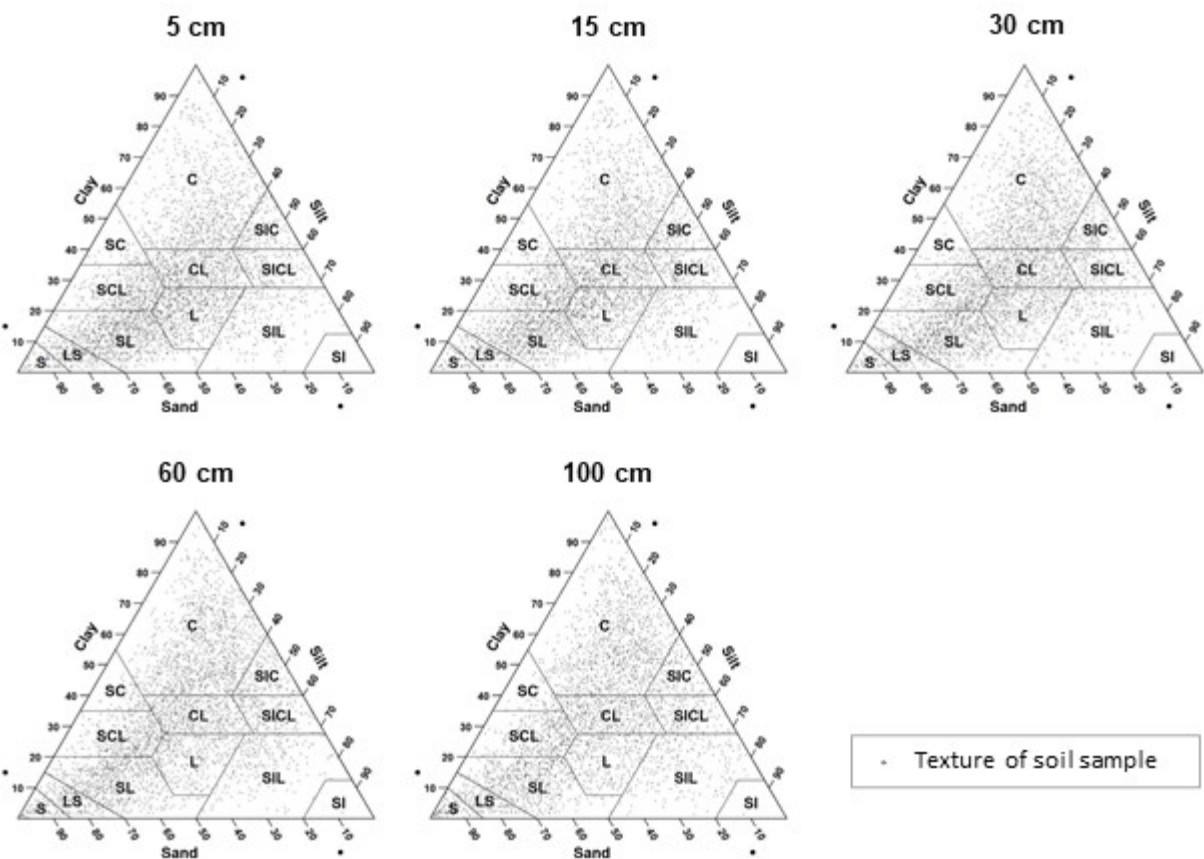

**Figure 2.** Particle-size soil samples representation in a textural diagram for each standard depth. C:Clay, SC: Sandy clay, SCL: Sandy clay loam, CL: Clay loam, SIC: Silt clay, SICL:Silt clay loam, L: Loam, SIL: Silt loam, S: Silt, SL: Sandy Loam, LS: Loamy sand and S: Sand.

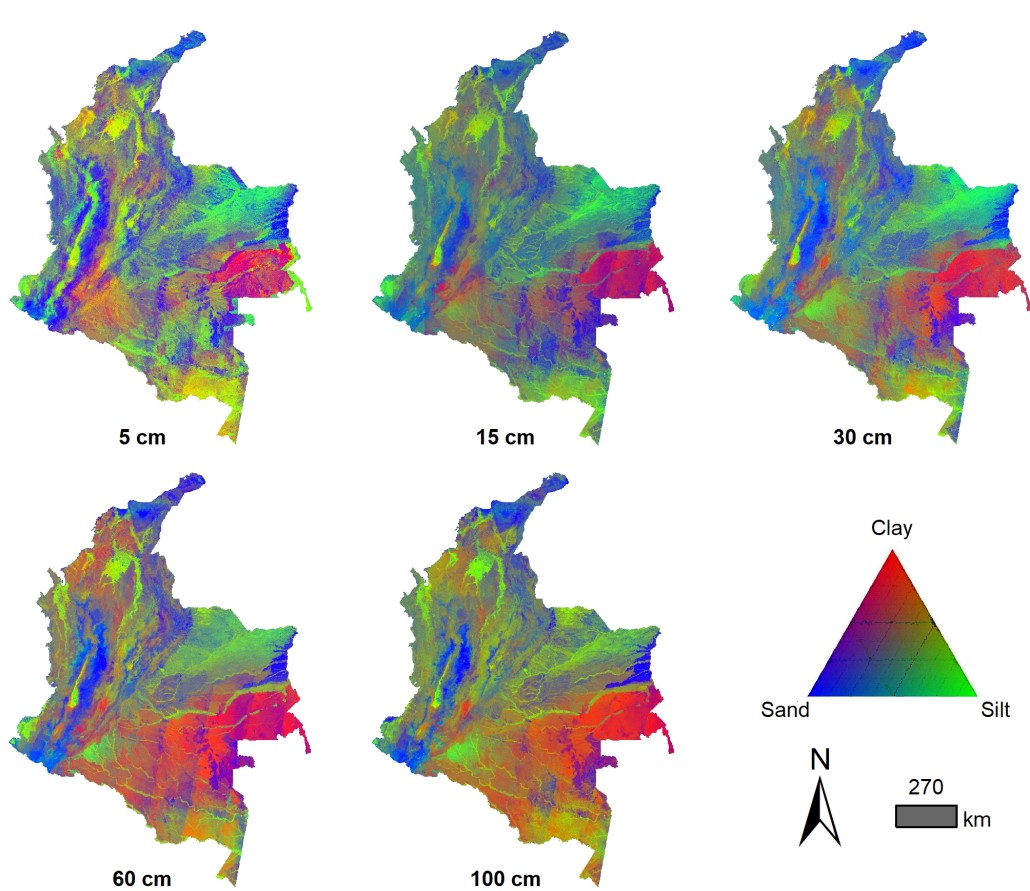

**Figure 3.** Color composite map of soil texture fractions predictions with the algorithm selected: landmap for 5 and 30 cm; MACHISPLIN for 15, 60 and 100 cm

**Figure 4.** Ensemble model, error distribution and best model selected at 5 cm

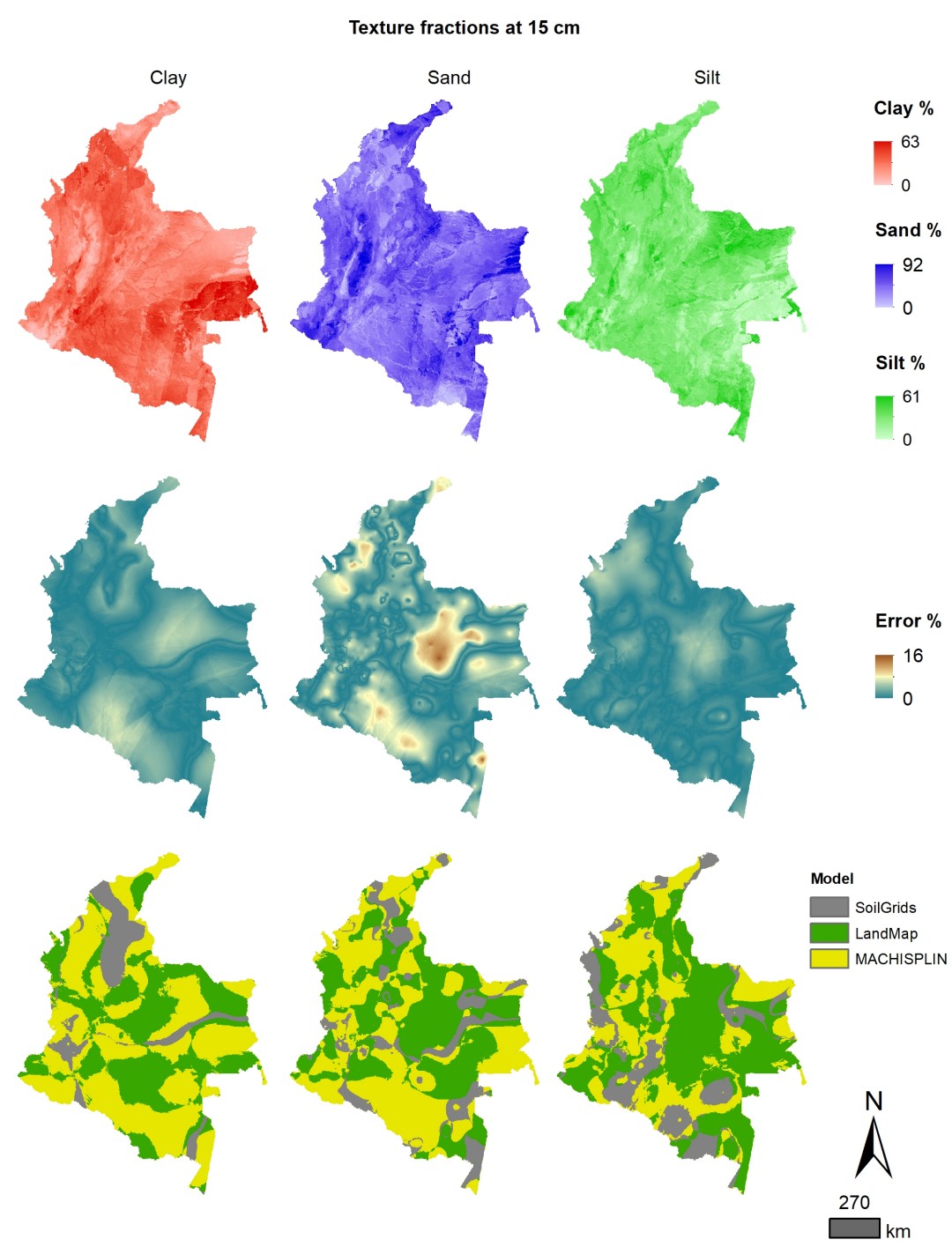

**Figure 5.** Ensemble model, error distribution and best model selected at 15 cm



**Figure 6.** Ensemble model, error distribution and best model selected at 30 cm



**Figure 7.** Ensemble model, error distribution and best model selected at 60 cm



**Figure 8.** Ensemble model, error distribution and best model selected at 100 cm