# Peer review of "Colombian soil texture: Building a spatial ensemble model"

_Earth System Science Data, 2021_

## Community Comment (CC1)

**General comments:**

The article deals with the generation of spatial maps of soil texture fractions at the country scale (Colombia) using an ensemble machine learning approach. The best explanatory covariates are selected for each transformation at all standard depths through a recursive feature elimination. The modeling process is explained descriptively and global SoilGrids products are also included in the validation process.

The article is well written with the introduction and discussion sections connected. However, it is far from a basic verification. As stated in the article, a logarithmic function was applied before modeling, and predictions were made with an ensemble algorithm such as MACHISPLIN. Then, % Clay, Silt, and Sand were obtained with a transformation function. If I'm missing something so far, please let me know if you reply.

You know, in the beginning, you checked a total of 100, considering consistency, in each layer (2.2 Data harmonization and transformation, Line 130). Well, I invite you to think about it by doing this on the raster maps you produce and putting your head between your hands. For this, for example, add three fractions at 5 cm multiples with any GIS program and raster calculation tool. Are the results in all 100? What does it mean in terms of uncertainty to see a distribution of this?

If you put the result of this in the article, you can find the opportunity to compare with Amirian- Chakan et al., (2019) and perhaps deepen your discussion. https://doi.org/10.1016/j.still.2019.06.006

The article will be carefully examined from different parts by many valuable anonymous reviewers. Verification with large-scale enterprise and global products is appreciated. However, making these processes more understandable is necessary for journal readers. The special issues I have mentioned below are related to writing.

**Specific comments:**

Line 37-38; Any soil scientist can use this phrase. So how do we measure it in the field? *with our hands*. However, the fractions are estimated separately for the study. In line with this, a new study that can be referenced to this sentence has been published very recently. You can take care of this. Use such a general sentence if you are interested in predicting classes of soil texture. We can't know how many % Clay by hand, we can separate the "Clay" class (or Clay Loam).

Related work: https://doi.org/10.1016/j.catena.2022.106155

Line 43-44; Are you the first to state that soil texture is costly, and the sampling process is time-consuming in large-scale studies? Please cite the source where you read this or could be an example.

Line 51; weathering or formation? Maybe, covariates can be surrogate soil formation factors. "Weathering" defines the higher deeply effective process through the profile.

Line 58-59; This is because soil texture has a compositional data structure. Clay, silt, and sand can be mapped separately in "continuous" data type, as well as spatial maps of the probability of occurrence of classes can be produced. You can show sources working probability maps of soil particle fractions and Categorical soil texture classes in continuous data type.

Related works: (Of course, you are not limited to these.)

https://doi.org/10.2136/sssaj2017.04.0122

https://doi.org/10.1016/j.geoderma.2016.09.019

https://doi.org/10.1007/978-3-030-85577-2_55

Line 67-68; Have these words been said before? for example;

https://doi.org/10.1111/ejss.13071

Line 75-80; Include references to the existence of this information.

Line 144 Table??

Line 150; Section 2.3. How did you generate these environmental variables? Which programs did you use? SAGA or QGIS with open access? In addition to these, did you use a commercial product? (ESRI, ArcGIS). Or it could be on a different program around the world.

Line 194; Section 2.5. Why did you resample after downloading the Soilgrids data in the validation part? Didn't you use the data in each depth layer at the locations where your current approximately 4000 data falls?

Are your actual values the ones in your dataset, and the predictions from the Soil grids? Of course, the Ml predictions are estimated with the ensemble model. Can you elaborate here? Because including the downloaded layers of soil grids in the "resample" process changes those values to a certain extent.

Line 225; Table ???

Line 235 Table ??? There are many like this in the article.

You stated that you have subtracted the estimations from the observed values for the error. It is given as % on the maps (Figure 4-8). Clarify

---

## Author Comment (AC1)

**Community comments**

**Fuat Kaya**

General comments:

The article deals with the generation of spatial maps of soil texture fractions at the country scale (Colombia) using an ensemble machine learning approach. The best explanatory covariates are selected for each transformation at all standard depths through a recursive feature elimination. The modeling process is explained descriptively and global SoilGrids products are also included in the validation process.

The article is well written with the introduction and discussion sections connected. However, it is far from a basic verification. As stated in the article, a logarithmic function was applied before modeling, and predictions were made with an ensemble algorithm such as MACHISPLIN. Then, % Clay, Silt, and Sand were obtained with a transformation function. If I'm missing something so far, please let me know if you reply.

You know, in the beginning, you checked a total of 100, considering consistency, in each layer (2.2 Data harmonization and transformation, Line 130). Well, I invite you to think about it by doing this on the raster maps you produce and putting your head between your hands. For this, for example, add three fractions at 5 cm multiples with any GIS program and raster calculation tool. Are the results in all 100? What does it mean in terms of uncertainty to see a distribution of this? If you put the result of this in the article, you can find the opportunity to compare with Amirian- Chakan et al., (2019) and perhaps deepen your discussion. https://doi.org/10.1016/j.still.2019.06.006

The article will be carefully examined from different parts by many valuable anonymous reviewers. Verification with large-scale enterprise and global products is appreciated. However, making these processes more understandable is necessary for journal readers. The special issues I have mentioned below are related to writing.

*Response: First, we would like to thank you for doing this review. We agree with you; the final product of this article needed to check with a "basic statement", which is that clay, sand, and silt percentages sum 100%. MACHISPLIN, landmap, and SG predictions fulfilled this statement; however, ensemble predictions did not. This comment was also in the referee's comments, and we needed to fix this issue. Each individual pixel was populated with a single model for all PSFs, and the model selection was made by calculating the Mean Absolute Error for the three PSFs.  We remade the figures 5 to 9 and we modified the methodology section.*

Specific comments:

Line 37-38; Any soil scientist can use this phrase. So how do we measure it in the field? with our hands. However, the fractions are estimated separately for the study. In line with this, a new study that can be referenced to this sentence has been published very recently. You can take care of this. Use such a general sentence if you are interested in predicting classes of soil texture. We can't know how many % Clay by hand, we can separate the "Clay" class (or Clay Loam). Related work: https://doi.org/10.1016/j.catena.2022.106155

*Response: We agree with you. It is necessary to clarify the identification of soil texture from analysis in the field or the laboratory. In the field, it is possible to identify texture classes (from our hands – organoleptic feelings), while in the laboratory it is possible to identify texture classes and particle size soil fraction (PSF) from different methods: Bouyoucos and pipette. However, we eliminate this part in the document because is too long, and it should be significantly shortened.*

Line 43-44; Are you the first to state that soil texture is costly, and the sampling process is time-consuming in large-scale studies? Please cite the source where you read this or could be an example.

*Response: No, we aren't. Several sources of information indicate that soil texture using standard laboratory analysis is costly and time-consuming in large-scale studies. Some authors are Bahrami et al. (2022) https://doi.org/10.1016/j.infrared.2022.104056, Jaconi et al. (2019) https://doi.org/10.1016/j.geoderma.2018.10.038, Kuang et al. (2012) https://doi.org/10.1016/B978-0-12-394275-3.00003-1, Azadnia et al. (2002) https://doi.org/10.1016/j.measurement.2021.110669, Kettler et al. (2001) https://doi.org/10.2136/sssaj2001.653849x. However, we eliminate this part in the document because is too long, and it should be significantly shortened.*

**Line 51; weathering or formation? Maybe, covariates can be surrogate soil formation factors. "Weathering" defines the higher deeply effective process through the profile.**

*Response: The correct word is formation. Let's change the word in the manuscript.*

**Line 58-59; This is because soil texture has a compositional data structure. Clay, silt, and sand can be mapped separately in "continuous" data type, as well as spatial maps of the probability of occurrence of classes can be produced. You can show sources working probability maps of soil particle fractions and Categorical soil texture classes in continuous data type.**

**Related works: (Of course, you are not limited to these.)**

https://doi.org/10.2136/sssaj2017.04.0122

https://doi.org/10.1016/j.geoderma.2016.09.019

https://doi.org/10.1007/978-3-030-85577-2_55

*Response: Yes, we need to show some sources of previous works that predict soil texture processing this soil property like a quantitative data type (percentages of clay, silt, and sand) and a qualitative data type (e.g., USDA texture classification). We modified the referenced sentence as follows:*

*"Predictions of quantitative soil properties (e.g., percentages of clay, silt and sand) (Liu et al., 2020; Li et al., 2020) or the probability of presence/absence of a soil class (e.g., a soil textural class) (Ramcharan et al., 2018; Kaya and Baˌsayiˇgit, 2022) are represented on digital soil maps for a given soil depth and for a specific period"*

**Line 67-68; Have these words been said before? for example; https://doi.org/10.1111/ejss.13071**

*Response: Yes, they have. In fact, the work developed by Wadoux et al. (2020) is a good example. They provided an introductory perspective on data-driven soil research by discussing some of the issues and opportunities of knowledge discovery from soil data. On the other hand, Khaledian and Miller (2020) (https://doi.org/10.1016/j.apm.2019.12.016) generated a study with the main considerations for working with model or algorithm, especially machine learning approaches. They highlighted that selecting the appropriate machine learning method depends on the digital soil mappers' purpose. Therefore, the emergent research question has been formulated by some research groups and other groups or they have begun to answer it. We will include these references in the manuscript.*

**Line 75-80; Include references to the existence of this information.**

*Response: We included references. Hengl and MacMillan (2019) https://soilmapper.org/, Geostatistics: Goovaerts (1997), Webster and Oliver (2007), and Haining et al. (2010) https://doi.org/10.1111/j.1538-4632.2009.00780.x.*
*Machine learning: definition -> Witten et al. (2011) https://doi.org/10.1016/B978-0-12-374856-0.00001-8.*
*Free models -> Wadoux et al. 2020 https://doi.org/10.1016/j.earscirev.2020.103359 and Kuhn and Johnson*

(2013) https://doi.org/10.1007/978-1-4614-6849-3. Cross validation or bootstrapping -> Brenning (2012) 10.1109/IGARSS.2012.6352393.

Line 144 Table??

*Response: All these edition mistakes were corrected in the document.*

Line 150; Section 2.3. How did you generate these environmental variables? Which programs did you use? SAGA or QGIS with open access? In addition to these, did you use a commercial product? (ESRI, ArcGIS). Or it could be on a different program around the world.

*Response: We generated the environmental covariables from several sources. They are indicated in Table 1. Except for the environmental covariates generated by Google Earth Engine (soil index and NDVI), the rest were generated using the geographic information system software ArcGIS version 10.3 (Esri, Redlands, CA, USA). We are going to include the commercial software in section 2.3.*

Line 194; Section 2.5. Why did you resample after downloading the Soilgrids data in the validation part? Didn't you use the data in each depth layer at the locations where your current approximately 4000 data falls? Are your actual values the ones in your dataset, and the predictions from the Soil grids? Of course, the MI predictions are estimated with the ensemble model. Can you elaborate here? Because including the downloaded layers of soil grids in the "resample" process changes those values to a certain extent.

*Response: We agree with you. We needed to do the validation step before the SG resampling. In this way, section 2.5 was rewritten and table 4 was updated. Additionally, comparing these two validation ways, the quantitative statistics are a little different, overall, about ME. This is a comparison for the first layers.*

| Depth | Map Quality measures | SoilGrids 250 m | | | SoilGrids 1 km | | |
|---|---|---|---|---|---|---|---|
| | | Clay | Sand | Silt | Clay | Sand | Silt |
| 0-5 cm | ME | 2.62 | -8.86 | 2.45 | 3.68 | -7.82 | 3.42 |
| | RMSE | 18.48 | 24.54 | 14.92 | 18.24 | 23.84 | 14.37 |
| | AVE | -0.10 | -0.14 | -0.13 | -0.07 | -0.07 | -0.05 |
| | CCC | 0.08 | 0.14 | 0.12 | 0.08 | 0.13 | 0.15 |
| 5-15 cm | ME | 2.16 | -8.35 | 1.73 | 3.47 | -7.15 | 2.86 |
| | RMSE | 18.41 | 23.93 | 14.55 | 17.89 | 22.94 | 13.85 |
| | AVE | -0.14 | -0.15 | -0.16 | -0.08 | -0.06 | -0.05 |
| | CCC | 0.06 | 0.13 | 0.12 | 0.08 | 0.13 | 0.15 |

Line 225; Table ???

*Response: All these edition mistakes were corrected in the document.*

Line 235 Table ??? There are many like this in the article.

*Response: All these edition mistakes were corrected in the document.*

You stated that you have subtracted the estimations from the observed values for the error. It is given as % on the maps (Figure 4-8). Clarify

*Response: The captions of the figures were clarified.*

**Referee comments**

**Referee 1**

General comments:

The manuscript "Colombian soil texture: Building a spatial ensemble model" by Varón-Ramírez et al. presents soil texture maps (clay, sand, and silt) for Colombia for different depth intervals by using and comparing different machine learning techniques. The authors compare their predicted maps with the global SoilGrid product for Colombia and provide maps that are based on the best model for each pixel. The soil data used to derive the maps and the final maps are provided as independent datasets/raster files and are easily accessible and usable. Unfortunately, the authors do not provide the code to reproduce their maps.

The manuscript and the related maps are unique and useful for the scientific community since, as stated by the authors, they provide the first Colombian soil texture maps obtained by spatial ensemble of national and global soil products. The methods are explained in detail (sometimes maybe too detailed for a general reader). However, I have quite a few comments regarding the manuscript and data presentation which I will outline below. Additionally, I think the language needs to be improved – sometimes the grammar and structure of sentences is not correct which makes it a little bit difficult to always know what the authors are trying to say. The usage of many abbreviations (which are mostly explained at the beginning of the manuscript) makes it also sometimes hard to follow the argumentation. Personally, I think it would help if you could just write out the words or use more intuitive acronyms.

*Response: The language was improved. Also, we fixed the acronyms and clarified them. We prepared a brief repository with our scripts (https://github.com/VimiVaron/Textural-maps-Colombia.git) (Version 1)*

Specific comments:

Abstract: Overall, I think the abstract is a little bit too long and detailed and should focus more on the novelty of the data products provided by this work and how they can be used. It is good that you describe what you did, however, it is probably not necessary to provide all the details about the model performance and comparison in the abstract.

*Response: The abstract was shortened and we emphasized the novelty.*

Line 5: How are the depth intervals exactly defined? I assume 0–5, 5–15, 15–30, 30–60, and 60–100 cm.

*Response: Yes, we are talking about these depths. The text was rewritten in the manuscript.*

Line 6: What do you mean by 'stack' in this context. Try to avoid to use too technical language in the abstract.

*Response: we refer to a "covariates compilation". The text was rewritten in the manuscript.*

Line 6: Maybe better: "the most important" instead of "top"

*Response: The text was rewritten in the manuscript.*

Line 10: Maybe better: "smallest" instead of "fewest"

*Response: Corrected*

Line 15: What do you mean by "compared to other algorithms"? Aren't all the methods you used spatial ensemble? Your abstract should really be understandable to a reader that is not that familiar with all the methods you used.

*Response: "compared to other algorithms" means "compared with EML and SG". The sentence was rewritten in the manuscript.*

Line 18: Should be "SPF" instead of "PSF".

*Response: We refer to soil particle-size fraction, we selected PSF as our acronym.*

Line 19: Without geographic context this information is difficult to follow. I think you can be less specific in the abstract and just say that the ensemble machine learning algorithms usually performed better, but in some regions the SoilGrid product also resulted in reliable predictions.

*Response: This information was removed.*

Introduction: The introduction gives a nice and comprehensive overview about the idea and state of the art of digital soil mapping in general and provides details about the methods used in this work. However, it can probably also be shortening a little bit. For example, the first two sentences (line: 29-31) are probably not needed.

*Response: We agree with you. We have shortened this section.*

Line 57: Maybe better: "Digital soil maps are derived from soil datasets that represent the continuous nature of soil variability."

*Response: We agree with you. We included 'derived from' in the sentence.*

Line 88: Missing brackets around references.

*Response: Corrected.*

Line 90: Not sure what you mean with "what are the best big-data management strategies for generating high-spatial resolution maps across large areas?". Your manuscript is not really about data management, but predicting soil maps.

*Response: In essence, our work is related to prediction and mapping. The data management item is part of the methodology chapter, but the emphasis is on prediction and mapping. The question was removed from this part.*

Line 101: Is your objective really to develop a digital soil texture dataset? As far as I understand, the soil data already exists and you are applying machine learning techniques to create digital soil maps. So, your objective should be more about the maps than the soil data. Maybe I am misunderstanding something here and you just have to be more clear what you did for this work and what is based on previous work.

*Response: We adjusted the paragraph to be clearer. In fact, the aim of the work is not to develop a digital soil texture dataset. Instead, we created a spatial ensemble model to produce maps of soil texture at different depths by applying machine learning techniques. However, we adjusted a dataset of soil texture (n=4203 at 0-5 cm) from several national surveys.*

Line 102: Are these soil data already part of any international soil databases (e.g. ISRIC, ISCN) so that they can also be used easy by other researches? I really encourage you to put your dataset in one of these international soil databases if you haven't done it yet.

*Response: Not exactly. This soil database is a new product of this work, and we make it available on Environmental Data Initiative (EDI), a service/platform to the scientific community that ensure environmental and ecological data are well curated and accessible for discovery and re-use well into the future. Also, EDI preserves environmental data for open and reproducible science. Therefore, the soil dataset is already part of an international soil database (not ISRIC or ISCN), and you can find it at: https://doi.org/10.6073/pasta/6dded07af834834ee21a134b247507fd.*

Line 117: Awkward phrasing: What do you mean with "positive implications"

*Response: We agree with you. In this case, we removed the expression "positive implications" because it generates confusion in the reader. We cleared the sentence.*

Methodology: This section provides a good overview about the applied methods. In general, I think that this section can be improved by focusing more on the actual methods rather than the R packages and functions that were used. If you provide the code to reproduce your maps this information can all be presented in the R script. Please also make sure that you always correctly cite the R packages that you are using – the reference is quite often missing.

Line 120: Awkward phrasing: "A total of five major steps". Maybe better: "Our work flow contains five major steps, including … which will be discussed in detail below".

*Response: We adjusted this part of the document.*

Line 124: Not needed: "Soil particle-size fractions (PSF) such as clay, sand, and silt were collected, including geographical coordinates (EPGS: 4326)." You can just write in the sentence before: "A total of 4,203 georeferenced (EPGS: 4326) soil profiles were collected from … that all contained information about particle-size fractions (clay, sand and silt)."

*Response: Corrected.*

Line 125: Did you create these geographic regions or are there somewhere defined. If so, please provide the reference.

*Response: No, we didn't create these geographic regions, they were defined previously by Rangel-Ch and Aguilar (1995). We included the reference.*

Line 130: Awkward phrasing of first sentence. Maybe better: "Dataset quality was ensured by i) sum of particle-size data equals 100 % and ii) no overlapping sampling depth". By definition, two soil horizons cannot be overlapping. Also, what about number of samples for each soil profile? Your Figure 1 shows that some profiles only have data down to 5 cm. Did you exclude any profiles that contained, e.g. less than 3 measurements or that did not reach a certain depth? If not, how did you treated these samples.

*Response: We adjusted the dataset quality section in the document. Regarding the number of profiles, or their exclusion, Figure 1 was changed and now it just shows the profiles at 0-5 cm (training and test) because it is easier for the reader. The total number of profiles was 4,203 at 0-5 cm, however, this number decreased with depth: 4201 at 5-15 cm, 4153 at 15-30 cm, 3974 at 30-60 cm, and 3597 at 60-100 cm. This means that the PSF predictions at 60-100 cm have fewer profiles than the PSF predictions at 0-5 cm, however, the treatment*

*was the same: 75% of the profiles at each depth were destined for training, while the rest were destined to testing. We included this part in the document.*

Line 131: "to" instead of "at"

*Response: Corrected.*

Line 133: Citation for R package aqp is missing

*Response: We include the citation. Beaudette et al., 2013.*

Line 136: Great that you are transforming the compositional data. However, I think you need to provide some more background information on why this is necessary and why you decided to use the additive log-ratio transformation (especially for non-experts in this field). It would also be helpful if you could provide some details on how to interpret the transformed values.

*Response: We rewrote a part of section 2.2 to include the explanation of the transformation and why this is important for PSF analysis. We use the ALR because is a widely used method for PSF analysis, however, we know that other transformations are possible; in this way we added a brief discussion about this.*

Line 142: Earlier you wrote, you applied the additive log-ratio transformation, now you are mentioning inverse additive log-ratio transformation. Please provide some more information what this is and why you are doing this. Also, in the published dataset it is not clear to me which of these transformations refers to transformation_1 and transformation_2. You need to provide this information in the metadata file and in the manuscript and maybe also think about a more self-explanatory naming of these two variables.

*Response: The inverse additive log-ration transformation is required to return to the original values after prediction. We rewrote this sentence. Additionally, we added a specification about Trans_1 and Trans_2 in the paragraph next to the equations 1 and 2.*

Line 146: Could you elaborate a little bit on why clay is used in the denominator? Just because it is used by other studies is not necessarily sufficient as an explanation.

*Response: our transformation (trans_1= ln(sand/clay) and trans_2=ln (silt/clay)) was selected in https://doi.org/10.1016/j.geodrs.2016.11.003 and https://doi.org/10.1016/j.catena.2020.104514. They made a brief verification testing other combinations (using sand as denominator and using silt as denominators); they found that using clay as the denominator, the transformations had a distribution close to normal distribution. We also made the verification (figure below) and we found that using clay as the denominator, the distributions obtained a smaller range, the standard deviations were so close, and the distributions were so like the other ALR transformations.*

[Figure]

Line 147: Missing citation for the R package Compositional.

*Response: We include the citation. Tsagris et al., 2022.*

Line 153: Table 1: Number of covariates does not match with the description of the covariates, e.g. soil has 28 covariates but only 5 are mentioned. A detailed list with precise data sources should be provided in the supplement. Also, the acronym GSI is not explained. What are the land categories you used? Are the extracted years matching with the year of sampling?

*Response: Table 1 shows a summary of the covariates employed. In the case of the column Soil formation factor 28 covariates are mentioned because soil index has 2 covariates, sand mineralogy has 10, clay mineralogy has 10, and moisture regime has 6. However, we adjusted the table with the complete covariates used in this work. GSI is a soil index (Grain Size Index). It is getting from Bands 2, 3, and 4 of Landsat 8 platform (Xiao et al., 2005). We used five categories from level 1 of Corine Land Cover Classification System (pastures, heterogeneous agriculture, shrubs, forest and permanent crops). However, the land cover study was carried out between 2010 and 2012, while the soil sampling included a broader range (before 2000 to 2015 approx.). In this kind of work, it is difficult to have the same period of land cover survey and the soil sampling because the soil sampling at a national scale has been developed from several projects, while the land cover survey is calculated mainly through remote sensing in a specific period.*

Line 154: Table ?? – missing cross-reference

*Response: We corrected all missing cross-reference.*

Line 154: How did you do the adjustment to 1 square kilometer? Could you provide some more details here, including uncertainties?

*Response: The covariates were of two kinds: binaries (Bin.) and continuous (Cont.). In the case of binary covariates, they were resampling in the R software using the function resample of the package raster. The method used to compute values for the new raster was "ngb" (nearest neighbor) due to the covariate features. In the case of continuous covariates, they were resampling in the same software, but the method used was "bilinear" (bilinear interpolation). DEM layer (derived of SRTM mission) was the raster with parameters that the covariates should be resampled. We didn't work with uncertainty covariates.*

**Line 155: Again, explain what you mean by "stack"**

*Response: A stack is a collection of rasters with the same spatial extent and resolution.*

Line 156: Missing citation for the R package caret

*Response: We include the citation. Kuhn et al., 2020.*

**Line 156 ff: The idea/method behind the recursive feature elimination is not clear to me. If I understand it correctly, you first built a model with all covariates and then selected the most important predictors (based on what?). How is it possible that you only then extract the values for the covariates at the profile level? Maybe, I am missing something here. So, if this a common approach it is probably fine, but I cannot really evaluate this.**

*Response: The recursive feature elimination (RFE) is an algorithm developed by Kuhn et al. (2022) in the caret package and implements backwards selection of predictors (covariates in our case) based on predictor importance ranking (http://free-cd.stat.unipd.it/web/packages/caret/caret.pdf). We generated a model for each depth and transformation. In overall there were 10 models, that is, we applied the RFE 10 times. Each model was analyzed with all covariates (83), but then with the RFE algorithm each model reduced the number of covariates. This algorithm operates by following these steps: First, the algorithm fits the model to all covariates. Then, calculate the performance of the model (RMSE, R squared, and MAE). The covariates are ranked, and the less important ones are sequentially eliminated prior to modeling. At each iteration of feature selection, the top-ranked covariates are retained, the model is refit, and performance is assessed. If the performance improves, the covariate is considered a valuable predictor. The final goal is to find a subset of covariates that can be used to produce an accurate model.*

**Line 166: Did you also consider different training and test datasets? In your introduction you mention spatial cross-validation which is probably a good approach for your data given its clustered nature. Could you please provide some more details about the bootstrapping technique since the splitting of the data is crucial for the validation of your predictions later.**

*Response: Yes, we considered different training and testing datasets. However, we were wrong to indicate the technique used. We didn't use the bootstrapping technique to split the dataset into training and testing. We used the createDataPartition function from caret package (Kuhn et al., 2020). This function generates a stratified random split of the data and aims to create balanced splits of the data. The sample is split into groups sections based on percentiles and sampling is done with these subgroups. We adjusted the manuscript with the correct process.*

**Line 165: Space missing**

*Response: Corrected.*

**Line 167: Space missing**

*Response: Corrected.*

Line 168–191: After reading this section I am not quite sure if I fully understand your methods. The description of the two R packages is quite technical and detailed, however, I am missing a little bit a more general description of the methods that are not necessarily restricted to the two R packages. Also, at the end of the section you mention spatial cross-validation, but earlier, you talked about bootstrapping the data. I think I am having a little bit difficulty to follow what you did at each step and why. Maybe you can emphasize this a little bit better. A flow chart might also help to guide the reader. Yet, I also have to admit that I am not really an expert in this field of digital soil mapping and other reviewers may be able to evaluate it better.

*Response:*

*About bootstrapping, we were wrong, this was also a comment from referee 2. We used the createdDataPartition function from caret package (Kuhn et al., 2020). This function generates a stratified random split of the data and aims to create balanced splits of the data. We rewrote the paragraph.*

Line 194: As mentioned by Fuat Kaya, why did you resample the SoilGrid data and not just extracted if for the sampling locations of your soil profiles?

*Response: We agree with you and Fuat Kaya, we needed to do the validation step before the SG resampling. In this way, this process was redone, the section 2.5 was rewritten, and table 4 was updated*

Line 195: Maybe better: "Next" instead of "After"

*Response: Corrected.*

Line 195: Did you do the validation for the SoilGrid product only or, as I assume, also for your own predictions? This is not really clear based on the sentence.

*Response: Yes, we did. In this work we validated three models: MACHISPLIN, landmap, and SoilGrids. The validation process used the testing dataset (25 % of samples). In the case of SoilGrids, we didn´t calculate the predictions, we just downloaded the SoilGrids layers and calculate the prediction error (difference between SG layer value and testing data). However, we adjusted the sentence for a clearer reading.*

Line 196–203: You do not provide an explanation/definition for the concordance correlation coefficient, but for all other measures you mentioned here. Additionally, the whole section (line: 194–203) is not that clear written: first you talk about comparing your results to the SoilGrid products and then you talk about the validation measures you used for your own predictions and the SoilGrid product. Again, try to streamline the description of your different steps so that it becomes more clear what you did when and why. I really think a flow chart-type figure would help here.

*Response: We provide more explanations about CCC. For the whole section (line: 194-203), we adjusted it to explain it clearly. In this case, we defined three steps (flow) to fully explain this section: i) layer adjustment, ii) calculation of quantitative statistics and iii) creation of prediction error maps. We think that with these changes the reader can get a clear idea of this section.*

Line 206: You did not introduce the term "independent residual" yet. Why did you not just used the prediction error terms? The interpolation was probably done for the final map – if so, say so here. Can you provide some more details about the kriging? There are a lot of different ways doing it.

*Response: We agree with you, the term "independent residual" is not necessary; it was removed.  Additionally, OK was used to map the prediction error for each model and each PSF. Next, this prediction error maps were used to calculate a mean absolute error (MEA) for each model, we explained this adding a figure 2 in the manuscript.*

**Line 210–115: Could you please provide some references for this section. Maybe you can also mention more clear that the spatial ensemble was based on your two models and the SoilGrid product.**

*Response: The paragraph was rewritten, and we added a flowchart of this ensemble process. Also, this flowchart was based on the suggestion of the referee 2, which purposed to do this ensemble calculating a mean error, this suggestion was to ensure the statement of soil texture (clay+silt+sand=100)*

**Results: Overall this section is quite descriptive (which is ok for a result section), however, I think the section could be improved by providing some more context and by focusing on the main results.**

*Response: The results were improved and rewritten.*

**Line 225: Table ?? – missing cross-reference; not clear which transformation refers to which.**

*Response: The missing cross-reference was corrected. The transformation refers to the Additive log-transformation, which is explained in section 2.2; additionally, to clarify, we added the application for equation 1 in the last paragraph of this section.*

**Line 225: Awkward phrasing: "the minimum contents were 1% or less for 5 standard depths" Maybe better: "For all depths, the particle-size fractions ranged from circa 0 to more than 90 %, except for silt, which only ranged from circa 0 to 80%". However, I think you can also just say that the particle-size fractions are covering more or less the entire range, which is to be expected for continental-scale analysis.**

*Response: The sentence was rewritten.*

**Line 234–236: This is repetition from the method section and probably not needed in the result section; Table ?? – cross-reference missing**

*Response: The sentence was removed.  The cross-reference was corrected.*

**Line 238: You have not defined the acronyms TEM, RH and PPT also units are not provided for the covariates. Did you scaled the covariates before using them? As I wrote earlier, you probably need to provide a table with all the covariates and description of them, including units and sources.**

*Response: Yes, we hadn´t defined the acronyms, only in the Table 3. Therefore, we included the acronyms in the body of the document. And yes, we scaled the covariates, especially according with their extensions, pixel resolution, and reference systems. We provide a table with all the covariates used.*

**Line 242: Table ??: missing cross-reference**

*Response: The cross-reference was corrected.*

**Line 254: Table ??: missing cross-reference**

*Response: The cross-reference was corrected.*

Line 262: Fig. 4–8 instead of listing all figures

*Response: The sentence was corrected.*

Line 265–274: It is not always easy to follow which model was best where, which is partly also due to wrong grammar. I encourage you to have someone checking the language and grammar throughout your manuscript. For example, what do you mean with "MACHISPLIN had representation in all natural regions, and in the deepest layers"?

*Response: We agree with you. We used a corrector to improve our language and grammar. We were taking about that MACHISPLIN predictions were selected in all regions in Colombia, also MACHISPLIN predictions were selected in the five standard depths. However, we rewrite this part that you are talking about in section 3.4*

Line 268: It should read MACHISPLIN instead of MACHISPLIS

*Response: The word was corrected.*

Line 275: Maybe better: "In terms of SG" instead of "Concerning SG"

*Response: The expression was improved.*

Line 278: Table ??: missing cross-reference; missing space

*The cross-reference was corrected and the missing space was added.*

Line 279: It should read "On the other hand"

*Response: The expression was improved.*

Discussion: This section provides some context for the results and also discusses limitations of the data and methods. However, it sometimes also repeats things from the introduction and result sections, which is probably unnecessary.

Line 286: "Soil texture is a key property required for many applications in environmental sciences" – This sentence is not adding anything new, and the statement was already made in the introduction.

*Response: We agree, the entire paragraph was rewritten.*

Line 288: Delete the second "previous"; what do you mean with the word "detail" in this context?

*Response: The second "previous" was deleted. With the word "detail" we refer to improve the resolution. We agree with you, this word is not clear, then the word was changed.*

Table 4: It should read "Root mean square error"

*Response: The expression was corrected.*

Table 5: Not clear what is meant by "adjusted parameters"; the table is showing the validation terms for clay, sand and silt for the five depth intervals. The acronyms are also not defined.

*Response: The expression was improved in Table 4 and 5, and the acronyms were defined.*

Line 290: Missing reference. Without any references it is difficult to follow your argumentation her. Also, this is probably material for the introduction and not really for the discussion.

*Response: the missing reference was corrected. The paragraph was rewritten.*

Line 290–296: This is a description of your methods and could probably be moved to the beginning of your result section to summarize what you did before presenting the results.

*Response: We agree, the paragraph was rewritten, and now, a part of these lines is in results section.*

Line 297: Does the soil diversity really changes with depth?

*Response: We needed to clarify this sentence. We were taking about soil properties (physical and chemical). The sentence was clarified.*

Line 297–312: This seems to be more part of the result section than the discussion section.

*Response: we disagree, we think that this paragraphs are part of the discussion, because we are describing the principal facts about PSF spatial distribution and the covariates related with.*

Line 327: "with" instead of "whit"

*Response: the word was corrected.*

Line 330: You are not providing any details why Araujo-Carrillo et al. 2021 provides the best example. I think in this section (line 323–343) you don't need to describe the methods of the other studies rather focus on comparing your results with their results and discussing which improvements you achieved and why a direct comparison is not that easy. You can then just briefly talk about differences and similarities at broad spatial scale.

*Response: We agree with you. It is not necessary to explain how the conventional maps in Colombia were made, then, we removed the line 323 to 329. However, lines 333 to 336 are necessary, because these limitations explained for conventional maps are the advantages of our results for Colombia.*

Line 343–355: I think this section could also be part of the introduction (and is partly already in it) as a justification for the methods you used in this work.

*Response: We removed this part from "discussion" to "introduction".*

Line 350: "than" instead of "that"

*Response: Corrected*

Line 356–369: Again, you provide a lot of details about the methods from other studies and yet, I don't think that this is necessary. In general, I like the idea of comparing error terms between the different studies, but I am not convinced that it needs to be done in such detail. Also, you have not mentioned this method (the error term comparison) anywhere else in the manuscript and it comes a little bit unexpected. So, I think you

either need to set it up better earlier in the manuscript or just provide a general discussion which study/method had better/worse error terms and what the reason for this is.

*Response: We removed this part from discussion, and we adapted some sentences in results section.*

Line 372: I think a word like "only" is missing between "but" and "in"

*Response: Corrected*

Line 374–376: Could you be more precise here? What do you mean with "in general terms" and "with good quantitative statistics"

*Response: The sentences were rewritten*

Line 383: Maybe better: "for the entire country of Colombia"

*Response: This part was rewritten*

Line 384: Sentence structure: "However, the differential factor included maps that represent the best model (EML or SG) in each area of the country at different depths, called in this work spatial ensembled."

*Response: This part was removed.*

Line 386: Unit % is missing; could you maybe discuss here what approach could overcome some of these limitations?

*Response: This part was removed.*

Line 389: What do you mean by "new and great challenges"

*Response: Maybe, 'new' is not a good word. We rewrite the sentence, adding other example of a great challenge.*

Line 391: Do you really think adding covariates will improve the predictions? You already tested 83 covariates. If you think you are missing important covariates, you should elaborate on why they are important and why they are missing.

*Response: In fact, we have used several covariates (83), however, most are binary covariates (62), while the rest are continuous (21). The binary covariates have been derivate from vectors (polygons in specific) of conventional studies (soil, geology, geomorphology, etc.) at coarser scales. Large areas in Colombia don't have at detailed cartographic scales to obtain better covariates. Another key point is related to all covariates are in fact 2D (i.e., values available at surface or for topsoil only), we copy the values of covariates for all depths in the regression matrix, which is a simplification. If we have covariates at different depths, we may obtain better relationships between covariates and PSF values (explain part of the variability).*

Line 392: "with" instead of "whit"

*Response: It was corrected.*

Line 396: What do you mean by "homosoil"?

*Response: This term was defined by (Mallavan et al., 2010) (10.1007/978-90-481-8863-5_12) and it refers to areas with homology of soil-forming factors. This includes climate, physiography, and parent materials. The homosoil identification is a quantitative methodology with the objective of being able to extrapolate a model from one area to another, where are not enough soil samples. This methodology has been applied by cited authors in the manuscript. The work of Mallavan et al. in 2010 was cited in the manuscript.*

Conclusions: The conclusion section many repeats statements from the result and discussion sections. Maybe think of some new aspects that could conclude the manuscript and only give a short summary of the main results and how they related to previous work.

Line 403: Is your map "just" better or does it reveal new patterns of soil texture in Columbia? Maybe you can add 1-2 sentence here (or earlier in the result/discussion section).

*Response: The maps are not only better than the existing soil texture maps provided by, for example, SoilGrids. The generated maps describe new patterns and distributions of PSF in several regions in Colombia. For example, in the case of sand, it's easy to identify higher percentages in the north (widely reported by the literature), however, this pattern is observed in the east of Orinoquia region, the southern of the Andean region, and in the middle of the inter-Andean valleys. We adjusted the conclusions, nevertheless, we kept the summary frame.*

Data availability: The provided links to the dataset and maps are working and everything can be downloaded easily. Please consider also to publish your code in order to fulfill the FAIR principles.

*We prepared a brief repository with our scripts (https://github.com/VimiVaron/Textural-maps-Colombia.git) (Version 1)*

Line 421: You have not clarified what trans_1 and trans_2 are and this information is also missing in the metadata of your published data.

*Response: to clarify what Trans_1 and Trans_2 are, we added the application of equation 2 in the section 2.2. Additionally, we updated our metadata.*

References: For some of the references the doi link is not correct, e.g. line 442

*Response: All references were updated.*

Figures:

Figure 1: Maybe think of using a different color scheme since red and green are not distinguishable for many people. www.colorbrewer2.org provides a nice tool for picking colorblind-friendly color schemes for maps.

*Response: We completely changed this figure. The new figure is clearer than the old one.*

Figure 2: Could you provide this figure in a better resolution?

*Response: the figure´s resolution was improved.*

Figure 3: Again, think of using a colorblind-friendly color scheme. Why are you not providing the maps for each model and each depth? It is not clear from the figure legend (which should stand for its own) why landmap is used for 5 and 30 cm and MACHISPLIN for the other depths in the same figure.

*Response: The colorblind RGB color is widely used for compositional maps (multilayer raster); it lets to appreciate the change of color for the most part of the people. Also, some studies used this colorblind to show compositional maps (for example, specifically in soil texture: 10.1016/j.geoderma.2019.114061).*

*On the other hand, we have shown the model with the best performance for each depth, as described in the figure´s legend; but you are correct, we need to show and compare the models, graphically. For this reason, we changed figure 4 in the manuscript.*

Figure 4–8: Colorblind-friendly scheme; The scale for clay, sand and silt should always have the same range, otherwise it is impossible to compare the maps which each other. Also, the contrast for the color range is not ideal, differences are difficult to see in the maps.

*Response: Figure 4-8 were corrected. We adjusted the color range.*

**Referee 2**

This is a very interesting study and I enjoyed reading it. The authors have applied some very novel ensemble machine learning algorithms to predict soil particle size fractions across the entire country of Colombia. The idea of also incorporating global predictions like SoilGrids with national model predictions is another novel application that can help to improve national digital soil maps in areas where training data is limited. The authors also treat the soil particle size fraction data as compositional data, using ALR transformed variables in their ensemble models. I think that this study has a lot of potential but there is one major issue that needs to be addressed.

Based on my understanding of the spatial ensemble approach presented here, this approach has some fundamental issues that need to be corrected. As correctly stated by the authors, soil particle size fraction data is compositional data. The authors appropriately used the ALR transformation for predicting the different PSFs. However, once the data is back transformed to sand, silt, and clay fractions, the authors then treat each fraction independently, thus ignoring their compositional nature. In their spatial ensemble, at a given pixel they may take the predicted clay from SoilGrids, sand from MACHISPLIN, and silt from Landmap, thus ignoring the interdependence of the predictions within a given model. From visually evaluating figures 4-8, it appears that this can result in pixels where modeled sand, silt, and clay percentages far exceed 100 percent. This observation was later confirmed in the manuscript on lns 385-387 and lns 410-411. Spatial ensembling of compositional data requires the preservation of the compositional structure of the data and the authors' current approach violates this. I see this as a fundamental flaw in the current approach that can be corrected.

I can see two possible solutions to this problem:

1. Each individual pixel would only be populated with a single model for all PSFs, e.g., all SoilGrids or all Landmap. Model selection would be based on the lowest model error averaged across the three PSFs.

2. Perform the spatial ensemble on the ALR transformed values (Tans_1 and Trans_2) for MACHISPLIN, Landmap, and SoilGrids. I'm not familiar with all of the processing steps in the MACHISPLIN algorithm but applying a similar approach for interpolating error and selecting the best model.

*Response: We agree with you, this was the major issue in our study. As we said in our introduction section, we needed to guarantee the preservation of the soil textural statement (soil texture is a compositional data). We followed your first suggestion: each individual pixel was populated with a single model for all PSFs, and the model selection was made by calculating the Mean Absolute Error for the three PSFs. We remade the figures 5 to 9 and we modified the methodology section.*

Another issue with the current spatial ensemble approach is that it produces lots of spatial artifacts (e.g., circle and blob patterns) which is likely due to the patchwork of models used to populate each pixel. Applying some type of model weighting, like the MACHISPLIN algorithm does, might improve these results. Another potential reason for the spatial artifacts seen in Figs 4-8 (e.g., vertical striping running north-south) is likely due to the ML algorithms. Documentation for the MACHISPLIN algorithm states that Boosted Regressive Trees and Random Forests models can produce blocky outputs, thus MACHISPLIN provides the option to exclude those two models from the ensemble using the model parameter 'smooth.outputs.only'.

*Response: We also agree with this point. In the revised version of the paper, we highlight and discuss the potential sources of these type of artifacts (lines 345 to 347 and 362 to 365). The value of this ensemble is that we can observe a 'most accurate' prediction for each pixel. The impact of using additional models or smoothing functions applied to the ensemble output should be carefully analyzed in future work. We provide all the means to reproduce our results and to compare and test multiple modeling approaches for identifying the optimal solution given user specific needs and areas of interest in Colombia.*

There are many more 'minor' issues that the authors should address which I list below:

Specific comments:

Introduction: The introduction is long and overly general. The authors should focus on topics directly relevant to this study, including ensemble modeling, spatial cross validation, and modeling compositional data. The section starting of line 59 extending to the sentence ending on line 96 does not add to the manuscript and can be replace with more relevant background text. For example this section provides a very general discussion of geostatistics which isn't direclty relevant to this study and discusses current research questions in DSM that are not addressed in this study.

*Response: We shortened the introduction section.*

Lns 60-61. Are there examples of unsupervised statistical learning for soil PSFs or texture classes? If not remove this statement.

*Response: We removed this part.*

Lns 100-101. Which accuracy indicators? The ones previously stated? If so, then say it rather than generally listing indicators and then not stating which ones you used.

*Response: We removed this part.*

Ln 123. From Fig. 1 it appears that not all sample locations were sampled at all depths. I assume this was due to the presence of shallow soils? If so please state this. Also, please provide a breakdown of the samples (training and validation) represented for each modeled depth. This could be easily included in Fig. 2.

*Response: We completely changed this figure. The new figure is clearer than the old one. It just indicates the samples for training and testing at 0-5 cm of depth. For the other depths, we don't include new figures because we don't want to generate lots of figures with similar trends, instead we indicate the samples (training and testing) at each depth in the document.*

Ln 124. 'Soil particle-size fractions (PFS)' -- acronym not consistent with abstract, i.e., SPF, soil particle fraction

*Response: We adjusted the acronyms. Soil particle-size fraction (PSF)*

Ln 145. Equations 1 and 2 need to be better defined, i.e., all equation parameters need to be explicitly defined. It is not clear in this particular application how Trans_1 and Trans_2 were calculated. You state that clay was used as the denominator variable, so does that make the denominator in equation 1 (zeta-D) equal to the clay fraction. Yet on line 141 you state that D =3, and i=3 (D) represents the silt fraction. Please explicitly define how Trans_1 and Trans_2 are calculated.

*Response: We clarified the application of the equation 2 in the methodology section.*

Table 1. Information on the spatial resolution of covariate data is missing for several sources, e.g., soil index, sand and clay minerology, landsat. It would also be helpful to provide an approximate grid cell resolution equivalent to the 1: 100,000 map scale. Also, many references in table are missing.

*Response: We didn't indicate the spatial resolution of some covariates because they are widely known, for example, Landsat 8 products (NDVI, band 6, or band 7 are worked at 30 m). However, in the case of the Soil Index (Clay ratio and GSI), we clarify in the manuscript how it is calculated and their inputs (essentially Landsat 8 products). It is necessary to indicate that all covariates were resampled to 1 square kilometer because it was our work resolution, according to the standard of Global Soil Partnership – GSP. In this work, we didn't calculate the right pixel size (a study developed by Hengl (2006) it's very interesting on this topic) because the spatial resolution was previously assigned (1 km$^2$). Sand and clay mineralogy layers were obtained from Soil Map by IGAC (2015). Table 1 spreads out the information about covariates.*

Ln 154 and throughout manuscript. Table citations are missing table numbers.

*Response: These edition mistakes were corrected in the document.*

Ln 154. 'adjusted to 1 square kilometer' Which upscaling method was used, e.g., nearest neighbor? Bilinear?

*Response: We used both. In the case of binary covariates, we used nearest neighbor. However, in the case of continuous we preferred bilinear interpolation.*

Lns 160-162. Please provide additional detials about this bootstrapping technique. Does it account for ranges in covariate space when splitting the samples? Was spatial autocorrelation accounted for when creating the training/testing split? Based on your statement on lns 189-190, without accounting for spatial autocorrelation, your training and testing datasets are not independent. on the other hand, there are arguments against the use of spatial cross validation, see  https://doi.org/10.1016/j.ecolmodel.2021.109692

Also, for reference, please provide the number of samples in training and validation sets, i.e., (75%, n=???)

*Response: We were wrong in these lines. We didn't use the bootstrapping technique to split the dataset into training and testing. We used the createdDataPartition function from caret package (Kuhn et al., 2020). This function generates a stratified random split of the data and aims to create balanced splits of the data. The sample is split into groups sections based on percentiles and sampling is done with these subgroups. According to the above subject, we didn't account for spatial autocorrelation to create the training/testing split. Spatial cross-validation is a feature of the landmap package, and we only indicated it. However, we included another reference against the use of spatial cross-validation (Wadoux et al., 2021). We adjusted the manuscript with the correct process. The number of samples for model training (75 %) and validation purposes (25 %) was refer to in line 141 in the manuscript.*

Lns 169-171. It would be good to provide additional details comparing the two ensemble modeling techniques. For example, the Landmap algorithm applies a stacking ensemble approach using 5 base learners and a 'meta model' or super learner to produce an ensembled prediction. What type of super learner was used? Also, how does the stacking ensemble compare to the weighting approach applied in the MACHISPLIN algorithm. How were the model weights calculated? These types of details are more relevent to this study than the very general discussion of machine learning vs geostatics presented in the introduction.

*Response: We modify the methodology section to explain these questions. The landmap algorithm trains 5 base learners and it construct a "meta-model" or "meta-learner" that is an additional model that basically combines all base learners. MACHISPLIN randomly makes a cross-validation while landmap makes the cross-validation doing a buffer to not select testing points that have spatial dependence. In general, in the first version of this manuscript, we tried to do an overview of digital soil mapping, because in Colombia are not may works related to this framework and we hope that our results will be used for stakeholders and governmental institutions; however, this idea resulted too long, we modified it for no distract the reader's attention, but we tried to maintain our intention. At final, as you and other reviewers suggested, we added some more context about our methods.*

Lns 171-174. This statement is not clear. Is the residual error interpolated for each model? Are these error surfaces used to determine the model weighting in the final ensemble? The details of how this is done needs to be made more clear.

*Response: The residual error is not interpolated for each model. The residual error is interpolated from the best model select with full dataset. The error surfaces are not used to determine the model weighing in the final ensemble. We completely adjusted and cleared this part of the document.*

Ln 187. How is the cross validation used to determine the meta-learner?

*Response: The answer about cross validation is derived from the work developed by Polley and van der Laan (2010). All details can be found at:*
*https://biostats.bepress.com/cgi/viewcontent.cgi?article=1269&context=ucbbiostat. A meta-leaner (L) is simply a collection of algorithms, and the cardinality of L is denoted as K(n). The algorithms may range from a simple linear regression model to a multi-step algorithm involving screening covariates. In the case of landmap package there are six algorithms. The steps followed by a meta-learner are a) Fit each algorithm in L on the entire data set. b) Split the data set (X) into a training (T(v)) and validation sample (V(v)), according to a V-fold cross-validation scheme. c) Fit each algorithm in L on T(v) and save the predictions on the corresponding validation data. d) Stack the predictions from each algorithm together to create a n by K matrix. e) Propose a family of weighted combinations of the candidate estimators indexed by weight vector α. f) Determine the α that minimizes the cross-validated risk of the candidate estimator over all allowed α-combinations. And finally, g) Combine α-minimized with the stack the predictions according to the family of weighted combinations to create the final super learner fit. In summary, the function can be expressed as the minimizer of the expected loss, and it is often the squared error loss. Although possible for the meta-learner to select only a single algorithm, it typically selects a weighted average of the algorithms together in an ensemble.*

Lns 189-191. The authors have done a nice job of citing recent work relevant to this study. In regards to the use of spatial cross validation, there has been recent debate as to its appropriateness for map validation. It might be good to reference this here.

https://doi.org/10.1038/s41467-020-18321-y

https://doi.org/10.1016/j.ecolmodel.2021.109692

*Response: We agree with you. We included these references in the manuscript.*

**Lns 194-195. So resampled from 250m to 1km? It is helpful to state this, as well as the resampling method.**

*Response: According to indicates above, we resampled several covariates to 1 km using two methods (nearest neighbor and bilinear interpolation). This part in the manuscript was adjusted.*

**Lns 206-207. Kriging assumes some spatial autocorreation among the errors. Was this the case? Might be helpful to provide the semivariograms. Did you consider using a thin-plate spline approach similar to the MACHISPLIN method?**

*Response: You are right, Kriging assumes some spatial autocorrelation. Attending your request, here we include a summary of variogram parameters. However, we consider that these do not have special relevance in the manuscript. Then, we added a brief sentence in the methodology section, where we explained that in each case the spatial autocorrelation was checked.*

| Depth | Model | PSF | Model | Nugget | Psill | Sill | Range | sserr |
|-------|-------|-----|-------|--------|-------|------|-------|-------|
| 0-5 cm | MACHISPLIN | Clay | ste | 14.815 | 200.948 | 215.763 | 38788 | 0.002 |
| | | Sand | ste | 169.741 | 171.265 | 341.006 | 121714 | 0.002 |
| | | Silt | ste | 34.350 | 112.211 | 146.561 | 142755 | 0.000 |
| | Landmap | Clay | ste | 142.370 | 86.116 | 228.486 | 47437 | 0.002 |
| | | Sand | ste | 214.110 | 156.787 | 370.897 | 212594 | 0.002 |
| | | Silt | sph | 132.341 | 50.215 | 182.557 | 32840 | 0.001 |
| | SG | Clay | ste | 166.670 | 136.059 | 302.729 | 84970 | 0.003 |
| | | Sand | Ste | 143.377 | 349.798 | 493.174 | 99723 | 0.002 |
| | | Silt | Ste | 73.495 | 116.820 | 190.315 | 65539 | 0.001 |
| 5-15 cm | MACHISPLIN | Clay | Ste | 181.866 | 13.514 | 195.381 | 188221 | 0.001 |
| | | Sand | Ste | 219.817 | 97.453 | 317.270 | 165708 | 0.001 |
| | | Silt | Gau | 108.749 | 15.217 | 123.966 | 105541 | 0.000 |
| | Landmap | Clay | Ste | 195.432 | 26.785 | 222.218 | 450490 | 0.006 |
| | | Sand | Ste | 207.225 | 154.590 | 361.815 | 153756 | 0.001 |
| | | Silt | Ste | 159.210 | 18.361 | 177.571 | 40688 | 0.000 |
| | SG | Clay | Exp | 215.976 | 96.461 | 312.436 | 237073 | 0.001 |
| | | Sand | Ste | 186.187 | 283.011 | 469.198 | 136422 | 0.003 |
| | | Silt | Ste | 118.079 | 60.804 | 178.884 | 154692 | 0.000 |
| 15-30 cm | MACHISPLIN | Clay | Ste | 222.545 | 15.480 | 238.025 | 112971 | 0.001 |
| | | Sand | Ste | 243.392 | 108.693 | 352.084 | 101495 | 0.001 |
| | | Silt | Ste | 67.859 | 69.118 | 136.977 | 84651 | 0.000 |
| | Landmap | Clay | Ste | 232.509 | 17696860 | 17697092.509 | 1732517143 | 0.001 |
| | | Sand | Gau | 282.905 | 58.798 | 341.704 | 71495 | 0.001 |
| | | Silt | Ste | 104.043 | 32.232 | 136.274 | 52046 | 0.000 |
| | SG | Clay | Ste | 238.269 | 102.557 | 340.826 | 473294 | 0.000 |
| | | Sand | Ste | 241.555 | 249.115 | 490.670 | 104466 | 0.002 |
| | | Silt | sph | 126.213 | 53.208 | 179.421 | 44632 | 0.000 |
| | MACHISPLIN | Clay | Ste | 229.132 | 63.382 | 292.515 | 156564 | 0.001 |

| 30-60 cm | | Sand | Ste | 197.278 | 230.033 | 427.311 | 58965 | 0.001 |
|---|---|---|---|---|---|---|---|---|
| | | Silt | Ste | 183.852 | 0.000 | 183.852 | 221565 | 0.001 |
| | Landmap | Clay | Ste | 244.931 | 44.364 | 289.296 | 188546 | 0.001 |
| | | Sand | Ste | 32.911 | 386.215 | 419.127 | 4897 | 0.002 |
| | | Silt | Ste | 173.155 | 16.832 | 189.987 | 607894 | 0.005 |
| | SG | Clay | Ste | 35.510 | 356.271 | 391.781 | 112549 | 0.002 |
| | | Sand | Exp | 0.000 | 521.749 | 521.749 | 4920 | 0.007 |
| | | Silt | Ste | 0.000 | 228.880 | 228.880 | 6271 | 0.001 |
| 60-100 cm | MACHISPLIN | Clay | Ste | 281.523 | 26.886 | 308.409 | 405216 | 0.008 |
| | | Sand | Ste | 455.243 | 64.971 | 520.214 | 400488 | 0.103 |
| | | Silt | Ste | 183.279 | 39.334 | 222.613 | 651916 | 0.033 |
| | Landmap | Clay | Ste | 295.021 | 23.078 | 318.098 | 367083 | 0.010 |
| | | Sand | Ste | 485.819 | 61.895 | 547.714 | 404714 | 0.087 |
| | | Silt | Ste | 190.667 | 49.427 | 240.094 | 561171 | 0.058 |
| | SG | Clay | Ste | 346.129 | 53.295 | 399.424 | 147813 | 0.002 |
| | | Sand | Ste | 571.456 | 39.254 | 610.710 | 201868 | 0.006 |
| | | Silt | Sph | 252.777 | 0.000 | 252.777 | 220783 | 0.003 |

Ste: Stein's parameterization; sph: Spheric model; Exp: exponential model; Gau: Gauss model.

**Lns 212-214.** The accuracy of this approach depends on the accuracy of your kriged error maps. It seems like applying a model weighting approach similar to MACHISPLIN might provide a better result rather than select the model with the lowest error at each pixel.

*Response: Yes, you are correct. However, the methodology planned for this study was to use ordinary Kriging to identify spatial distribution of the errors. However, this is a great recommendation to improve this work.*

**Ln 242.** 'Boundary adjustment parameters'? I'm not sure what you mean by this. It is not referenced anywhere else. Do you mean Accuracy metrics or indices?

*Response: we corrected the expression, we changed this for "map quality measures".*

**Tables 4 and 5.** 'Adjustment parameters'? Why are these model accuracy metrics referenced as adjustment parameters? Also, it should be stated here that these accuracy statistics are based on the validation dataset.

*Response: we recognize that this term is not clear. We changed this for "map quality measures". We corrected this in the tables.*

**Table 5.** Why was CC for clay at 5cm lower than either MACHISPLIN or Landmap for that depth and fraction? I would have thought the spatial ensemble would select the most accurate model for each site and therefor produce more accurate results relative to the other models. There are other instances of this among the depths and fractions. Could this be a result of combining PSFs from different models?

*Response: With your main suggestion to improve our ensemble maps, we remade the maps. With this new methodology, the RMSE, AVE, and CCC were improved or, in the worst case, these were equal to either MACHISPLIN or Landmap. However, ME in some cases increased, then the predictions were a few biased compared to either MACHISPLIN or Landmap. Also we updated tables 4 and 5.*

Ln 292. Is this a reference to the predicted map uncertainty for SoilGrids? Was this uncertainty evaluated? It would be interesting to see uncertainty maps of SoilGrids PSFs for this area. This is also an important aspect of digital soil mapping not addressed in this paper. Since SoilGrids quantifies model uncertainty this would be an interesting point of comparison to national model results.

*Respond: We didn't evaluate the SoilGrids' uncertainty maps. The reason was to apply the same methodology and scheme to calculate the error in all the methods evaluated. The SoilGrids uncertainty maps were computed from the global soil database, not from a national (local) soil database like the one related in this work. This difference is very important for us because we need to identify the best method and layer in each PSF in Colombia using national data.*

Lns 299-301. Fig 2. is presented in black and white and as a low resolution figure. making it difficult to interpret.

*Response: this figure was remade, and we added some colors.*

Lns 385-387. Was the data normalized to 100 after the spatial ensemble? This might explain why some of the PSF at certain depths had lower performance relative to the EML models. I see this as a major problem with your spatial ensemble approach.

*Response: We applied your first suggestion to fix this problem. Then, in our final ensemble maps, we assure the sum of PSF are equal to 100% in the entire area.*

Lns 402-403. This was seen in Table 4 with the validation statistics. What is missing is a visual comparison of the two EML algorithms. In Fig. 3 you show either Landmap or MACHISPLIN but not both. I would like to see a figure similar to Fig. 3 but showing Landmap, MACHISPLIN, SoilGrids, and the spatial ensemble at one or two depths.

*Response: We agree with you. We took advantage of the comparison, and we remade this figure as you suggested.*

Lns 410-411. Evaluating model accuracy in these areas is tricky because there is limited data to accurately model the spatial distribution of model error. Using ordinary kriging won't do a great job in these data sparse regions.

*Response: We agree with you. At all depths in the south area, the smallest amounts of soil samples in the country are reported. This situation generates more uncertainty (or error) in these areas. However, with SoilGrids products, the results are slightly better than the other methods.*

Lns 414-416. It is good to see that the authors recognize this issue with the spatial ensemble approach. However, I see this as a major flaw that diminishes or even removes the prior efforts to account for the compositional nature of the data. This could have been avoided using one of the alternative approach I outlined above.

*Response: As you suggested, we fixed this issue whit our ensemble maps.*

**Referee 3**

Soil texture is one of fundamental soil properties. Obtaining better estimation of soil texture for sites without samples is the key for the generation of soil texture maps. The study presented texture maps for Colombia at five standard depths (5, 15, 30, 60, and 100 cm) obtained via spatial ensemble of national and global digital

soil mapping products. A comparison of newly developed soil texture maps with the previous maps (SoilGrids, SG) showed the improvements of new maps. The datasets shared in the study are valuable.

- What are differences between "SPF" and "PSF"? Line 18, What is "PSF"?

*Response: There is not any difference. We make a mistake. The correct expression is PSF (soil particle-size fractions).*

- A thorough reading is suggested. It seems that there are some missing information. Such as Line 154 Table ??.....

*Response: All these edition mistakes were corrected in the document, especially the table labels.*

- Table 3, it was showed that top five Covs were selected. Why five Covs? How many explained variances by five Covs?

*Response: We just indicated the selection of the top 5 covariates for each transformation and standard depth because we wanted to express the main results of section 3.2 (covariate selection). In this section ten models with ranges between 44 and 83 were calculated. We didn't just use the top 5 covariates; we used all the covariates generated by the recursive feature elimination algorithm (rfe) for each transformation and standard depth. The rfe function easily produces the top 5 covariates in its results. However, in Table 3 we summarized some results of this section. The rfe function implements a backwards selection of predictors based on predictor importance ranking. The goal is to find a subset of predictors that can be used to produce an accurate model. The algorithm starts by fitting a model using all covariates, assessing its performance, and ranking covariate importance. The least important covariates are then removed from the pool, and again the model is fitted and assessed, and the least important covariates are removed. In this case, we didn't use the explained variance, we used another summary metric (RMSE) to select the optimal model.*

**Referee 4**

This study used digital soil mapping framework to generate the first texture maps at five depths of Colombia. They used additive log-ratio (ALR) transformation on sand, silt, clay content to develop models. Two ensemble machine learning methods (MACHISPLIN and landmap) and predicted maps from SoilGrids were compared and a spatial ensemble function was created to select the best model for each pixel in the final maps.

I have some specific questions:

Introduction: The introduction is too long, and it should be significantly shortened. Some general sentences can be removed, for example, lines 97–101, Models or algorithms for digital soil mapping are evaluated …

*Response: We shortened the introduction section. We removed some information which no added information for our study.*

The language should be improved. Some sentences are incorrect. For example, line 109, "Understanding which are the prediction algorithms and approaches yield lower error levels at the pixel level…" delete "are the".

*Response: We agree with you. We have shortened this section. Also, we adjusted some incorrect sentences.*

Methodology: This section is clear and easy to follow, but more details should be provided.

Line 123, how did you sample the profiles? Based on identified horizons in the field?

*Response: All the profiles were obtained from SISLAC. Each profile was evaluated following the section 2.2. In SISLAC the horizons were identified in the field.*

Line 124, what method was used to measure the clay, sand, and silt in the laboratory?

*Response: SISLAC don't report the laboratory method used for each profile. However, the main source in Colombia is IGAC and uses Bouyoucos and pipette in the laboratory.*

Line 131, five standard depths (5, 15, 30, 60, and 100 cm). I think it is better to change it to 0–5, 5–15, 15–30, 30–60, 60–100 cm throughout the paper.

*Response: Corrected.*

Line 146, why was clay used as the denominator? Is there any difference if you use sand or silt?

*Response: our transformation (trans_1= ln(sand/clay) and trans_2=ln (silt/clay)) was selected in https://doi.org/10.1016/j.geodrs.2016.11.003 and https://doi.org/10.1016/j.catena.2020.104514. They made a brief verification testing other combinations (using sand as denominator and using silt as denominators); they found that using clay as the denominator, the transformations had a distribution close to normal distribution. We also made the verification (figure below) and we found that using clay as the denominator, the distributions obtained the smaller range, the standard deviations were so close, and the distributions were so like the other ALR transformations.*

Line 150, 83 environmental covariates, please provide information on how you obtained these covariates and description of these covariates. For example, there are 6 lithology and 10 soil orders, what are these? What are the 5 oblique geographic coordinates? How did you adjust the covariates to 1 square kilometer? Which resample method did you use?

*Response: We include a supplementary material with the complete covariates used in the work and your description. Lithology and soil orders were binary covariates generated from national soil survey (IGAC (2015)). The oblique geographic coordinates were covariates generated from OGC package version 1.0.1 developed by Møller et al. (2020). This package creates several tilted coordinate rasters. We used the function resample from the raster package to adjust the covariates to 1 square kilometer. According to the type of covariate, we used the nearest neighbor in the case of binary covariates, and we preferred bilinear interpolation in the case of continuous.*

Line 154, Table ??. None of the Tables are clearly mentioned in the text.

*Response: this was corrected in the manuscript.*

Line 194, add the original resolution of SG and which method did you for the resample? Do you know roughly how many profiles the SG used in Colombian area?

*Response: Corrected. SoilGrids reports 236 profiles:*

[Figure]

Line 195, "external validation". I think it's more common to say "independent validation".

*Response: Corrected.*

Line 200, is AVE the same as R2 (coefficient of determination)?

*Response: Yes, it is. In the case of linear regressions, both concepts are the same. R2 is the amount of variation explained by the model, and AVE (Amount of Variance Explained) is also known as coefficient of determination or ratio of scatter.*

Line 206, so the error maps in Figs. 4, 5, 6, 7, 8 are from interpolation of residuals using ordinary kriging. In a widely used method – regression kriging, the final map is obtained by adding up the regression map and residual kriging map. Have the authors considered doing the same thing? In addition, the error map is not regarded as uncertainty map. Have you considered calculating model uncertainty using other methods, e.g., bootstrapping?

*Response: We use ordinary Kriging for all spatial distribution of errors. We based our ensemble maps in the lowest error at each pixel and we use ordinary kriging to identify this distribution because this is the most widely used geostatistical method to estimate a continuous variable in unsampled locations. For the following works, we could explore the best way to identify this error distribution, maybe as reviewer 2 proposes (using a thin-plate spline approach), and how this could improve our actual results.*

Results:

Line 250, it should be "decreased with increasing depth". Similarly, line 257, "the RMSE increased with increasing depth".

*Response: the sentences were rewritten.*

Table 4 and 5, why do you use "adjustment parameters"?

*Response: we corrected the expression, we changed this for "map quality measures".*

Discussion:

Line 288, "While many studies focus on mapping soil properties such as pH and organic matter, less studies focus on comparing and testing approaches for maximizing accuracy." I think there are many papers focusing on method development and improving the accuracy.

*Response: we agree, there are many works focusing on development and improving accuracy. But the special topic in this work is to test global approaches, like SG, to improve the accuracy. Then, the sentence was clarified.*

Line 298, "As depth increases, the soil thins, and the proportion of clay and silt rises.". Unclear sentence. What does "soil thins" mean?

*Response: We referred that the soil texture in the deepest layers is finer than in the superficial layers. The sentence was rewrite in the manuscript.*

Line 380, Fig. 1 is first cited in the discussion. It should be mentioned before Fig. 2.

*Response: The citation was removed.*

---

## Author Response (AR2)

**Reviewers' comments (Second round)**

**Reviewer #1**

Some minor issues:

Line 11, "their accuracies were very similar to the PSF at each standard depth". I think it should be either "their accuracies were very similar at each standard depth" or "their predicted values were very similar to the measured PSF at each standard depth". You can't say accuracies are similar to PSF

*Response: You are correct. We improved the writing of this sentence.*

Line 126, use "predicted results" instead of "predictive results"

*Response: the sentence was corrected.*

Figure 2, I think the framework is a little confusing. The input maps, selection criteria and procedure, and final maps should be better distinguished from each other by different shapes, and the flow should be clearer. Here, it seems the parallelograms are used for maps, but it is also used for "Min MAE" and "Model selected". While the "Min MAE" should be the criterion to "select Model".

*Response: We agree. The flowchart was a little confusing. We made the figure again and used a different color for inputs and outputs. We restructured figure 2, and we specified the criteria for each process.*

**Reviewer #2**

General comments:

I acknowledge that the authors have heavily revised the manuscript and have addressed most of my previous comments. The manuscript has improved significantly from the previous version. I only have a few minor comments – mainly clarification. Beside this, the grammar and sentence structure are still partially incorrect and sometimes hinder the flow of reading. Therefore, I recommend a careful language edit before final publication.

*Response:*

Specific comments:

Line 9f: I think there are already national texture maps based on digital soil mapping. You could highlight that this is the first national texture map based on digital soil mapping in Colombia.

*Response: We agree. The sentence was rewritten to clarify the statement.*

**Line 28f: I disagree that clay content always has to increase with depth. Although this is quite common, it is not a requirement for mapping – not like that all three texture classes always have to sum up to 100%.**

*Response: We agree. Clay content not necessarily increases with depth. We wanted to say that PSF could variate in proportion while depth increasing. We rewrote the sentence like this:*

*"Additionally, the soil texture study must include two principal statements: first, it is compositional data, which means that PSF sum to 100% (%clay+%sand+%silt), and this statement must be satisfied at each location (Amirian-Chakan et al., 2019); second, the proportion of PSF could variate between horizons depending on soil forming factors interactions (Orton et al., 2016; Poggio and Gimona, 2017)."*

**Line 43: Does not every statistical model needs a response variable?**

*Response: We can find two principal categories of statistical learning: supervised and unsupervised. In supervised learning, for each observation of the predictor measurement, there is an associated response measurement (response variable). While unsupervised learning describes the somewhat more challenging situation in which, for every observation, we observe a vector of measurements but no associated response; in this case, we can seek to understand the relationships between the variables or between the observations (James et al., 2013). Some tools for unsupervised statistical learning are cluster analysis and principal components analysis. Some applications of unsupervised statistical learning in digital soil mapping could be 10.1016/j.geoderma.2021.115012 and 10.1016/j.geomorph.2020.107305.*

*We rewrote the sentence to clarify the general statement:*

*"These predictions or probability estimates are derived from the use of supervised statistical learning (in the presence of training data for a response variable) or unsupervised statistical learning (in the absence of a response variable) (James et al., 2013)."*

**Line 73: What does "RMSE ≈ 122,200" mean?**

*Response: We agree; the sentence is not clear. 122.200 are the number of points or observations. We rewrote the sentence to clarify the statement.*

*"One representative case was developed by Hengl et al. (2021) for the continent of Africa at three depths (0, 20, and 50 cm) and 30 m spatial resolution. They produced predictions using two scale 3D ensemble machine learning (EML) framework and 122,200 training samples (approximately); their study utilized an improved predictive mapping framework: spatially-adjusted EML, that better accounts for spatial clustering of points. The spatial cross-validation methodology was a special point of their work, obtaining RMSE values of 9.6\%, 13.7\%, and 8.9\% for clay, sand, and silt, respectively."*

**Line 98: Could you provide a short description of the soil texture classes? How are they defined in terms of size (e.g. silt: 0.002 to 0.05 or 0.002 to 0.063 mm).**

*Response: We added a brief description for soil particle-size classification. The textural classification in the Soil survey studies in Colombia is made according to the United States Department of Agriculture (USDA). We rewrote the sentence as follows:*

*"A total of 4,203 georeferenced (EPSG:4326) soil profiles were collected from Sistema de Información de Suelos de Latinoamérica y el Caribe - SISLAC, a soil information system developed by FAO \citep{FAO:2020:Online}, that all contained information about soil particle-size fractions (PSF). These PSF are classified according to the United States Department of Agriculture (USDA) system; clay (particles smaller than 0.002 mm in diameter), silt (particles sizes from 0.002 to 0.05 mm in diameter), and sand (particle sizes from 0.05 to 2.0 mm in diameter)."*

**Line 162: Only a side-comment. The package mlr is not maintained anymore and the most recent version of this package is called mlr3 (https://mlr3.mlr-org.com/).**

*Response: Effectively, package mlr is considered retired from the mlr-org team. They suggest to use the new mlr3 framework from now on and for future projects. However, the creator of landmap package indicated that landmap provides methodology for automated mapping using Ensemble Machine Learning extends functionality of the mlr package (Hengl, T.:2021. landmap, url:https://github.com/envirometrix/landmap). Nevertheless, we adjusted the name of the new package.*

**Line 202: See my comment about (line 9).**

*Response: The sentence was rewritten as follows:*

*"This study represents the first effort to provide a national map of soil texture using a digital soil mapping framework in Colombia."*

**Line 217: Figure 2: I find the flow chart really helpful. However, I was wondering why you show PSF error as an input into the models? Did you not use the transformed PSF values?**

*Response: In the spatial ensemble procedure, the inputs are the PSF predictions (MACHISPLIN, landmap, and SG), and the selection is based on the minimum MAE (previously calculated). We made this figure again.*

**Line 259: This should read Figure 5 to 9**

*Response: Corrected*

**Line 266: What do you mean by "extensive and continuous areas"?**

*Response: We agree with you: the term is not clear. We made a mistake with this expression because Amazon region is the largest in Colombia, but Orinoquia is not, and neither is continuous. We rewrote the sentences including the characteristics of MACHISPLIN predictions.*

**Line 273: What do you mean with the acronym EM? Should it read ME? But then the sentence does not make sense.**

*Response: You are right, the acronym was incomplete. It should be EML (Ensemble Machine Learning), as was defined in line 147 (methodology section): MACHISPLIN and landmap. The sentence was corrected as follows:*

*"The ME values were closer to zero, showing an improvement in the prediction; however, in this ensemble model, predictions of silt fraction had the highest bias, which is a different behavior of EML algorithms (MACHISPLIN and landmap), where sand fraction had the most biased predictions"*

**Line 373: In line 273 you wrote: "RMSE values decreased for all PSF and standard depths, which means a raising in the precision of the map." And here you write: "However, layers of 0-5, 5-15, and 15-30 cm obtained the best results". Maybe I missed something, but to me this is contradictory.**

*Response: In line 273 we were taking about the comparison between spatial ensemble maps and EML or SG products maps. While, in line 373 we are referring only for EML maps. To clarify this, we rewrote the first sentence in line 273 follows:*

*"The external validation for the spatial ensemble maps showed an improvement in their metrics vs. the maps using EML algorithm or SG product (Table 5)"*

**Line 392: I really appreciate it that the authors provide the R code for this work.**

*Response: We recognize that provide the R code is necessary and helpful.*